# Spatial and temporal variability of sea surface temperatures and monsoon dynamics in the northwestern Arabian Sea during the last 43 kyr

Jan Maier[1,2], Nicole Burdanowitz[1,3], Gerhard Schmiedl[1,3], Birgit Gaye[1,3]

[1] Institute for Geology, Universität Hamburg, Bundesstraße 55, 20146 Hamburg, Germany

[2] Friedrich-Alexander-Universität Erlangen-Nürnberg (FAU), Department Geographie und Geowissenschaften; 91054, Erlangen, GeoZentrum Nordbayern, Schlossgarten 5, Germany

[3] Center for Earth System Research and Sustainability (CEN), Universität Hamburg, Bundesstraße 55, 20146 Hamburg, Germany

*Correspondence to*: Jan Maier (jan.m.maier@fau.de)

## Abstract

In this study, we present the first well-dated, high-resolution alkenone-based sea surface temperature (SST) record from the northeastern Oman Margin (Gulf of Oman) in the northwestern Arabian Sea. The SST reconstructions from core SL167 span the last 43 kyr and reveal temperature fluctuations of around 7 °C (ranging from 20.1 °C to 27.4 °C). Thus, this region has a higher sensitivity to climate variations compared to other core locations in the Arabian Sea and fills a gap in a previously unstudied region. SSTs were lowest during Heinrich event 4 (H4) and were comparatively low during H3, H2, the Younger Dryas, early and late Holocene. Comparatively higher SST occurred during some Dansgaard-Oeschger interstadials (D-O 11, D-O 4 – 9), the Bølling-Allerød (B-A), and the mid-Holocene. The SST was predominantly influenced by the SW monsoon during warmer periods and the NE monsoon during cold intervals. Importantly, the last glacial maximum stands out by absence of intense cooling at the core site, which clearly diverges from previously known SST patterns. We speculate that this pattern was caused by stronger NW winds and an eastward shift of the SST gradient in the Gulf of Oman, resulting in a brief and moderate cooling period. Strong SW winds during the early Holocene transported cold water masses from Oman upwelling into the Gulf of Oman, lowering SSTs. A rapid temperature increase of approx. 2 °C during the mid-Holocene was likely induced by the weakening of SW winds and an abrupt eastward shift of the SST gradient.

## 1 Introduction

The Arabian Sea is impacted by one of the world's largest and most complex climate systems – the Indian monsoon system (Gupta et al., 2020). Seasonal monsoon winds (Figure 1) are driven by alternating atmospheric pressure gradients and induce regional and annual fluctuations in sea surface temperature (SST) patterns (Figure 2a-d). The SW monsoon significantly influences precipitation patterns in the monsoon region, accounting for approximately 80 % of the total modern annual

precipitation (Gadgil, 2003). This monsoon system also dictates environmental conditions, affecting phenomena such as droughts, floods, and terrestrial vegetation coverage. Moreover, it plays a crucial role in shaping the economies and societies of southern Asia and the Arabian Peninsula (Clift and Plumb, 2008; Krishna Kumar et al., 2004). The strength of the monsoon has undergone shifts and has changed due to past global climate variability. These monsoon fluctuations are associated with

significant hydrological changes, including alternating phases of excessive and deficient precipitation, causing severe challenges for civilizations in the region (Gadgil, 2003; Krishna Kumar et al., 2004). The Indian monsoon system is an important component of the global climate system and sustains the livelihoods of over a billion people worldwide (Gupta et al., 2020). Therefore, utilizing high-resolution paleo-reconstructions is essential to enhance our understanding of forcing mechanisms and their impact on monsoon variability in the past, as well as to improve forecasting and future climate modeling.

The last glacial period was characterized by significant climate variability in the northern North Atlantic, known as Dansgaard-Oeschger (D-O) oscillations and Heinrich Events (Bond et al., 1993; Dansgaard et al., 1993; Heinrich, 1988; Johnsen et al., 1992). These oscillations are linked to instabilities in the Northern Hemisphere glacial ice-sheets, resulting in a significant freshwater influx into the North Atlantic. This, in turn, impacts the Atlantic Meridional Overturning Circulation (AMOC) and can lead to a substantial reduction or even a complete shutdown (Broecker, 1994; Dansgaard et al., 1993;

Ganopolski and Rahmstorf, 2001; Heinrich, 1988; Hemming, 2004; McManus et al., 2004). Building upon these insights, previous studies (e.g., Leuschner and Sirocko, 2000; Reichart et al., 1998; Sirocko et al., 1996; Schulte et al., 1999; Schulz et al., 1998) have described a strong connection between the North Atlantic Ocean and the Indian monsoon climate in the Arabian Sea during stadial (Heinrich Events, Last Glacial Maximum) and interstadial periods (D-O cycles, Bølling-Allerød (B-A), Holocene). To enhance our understanding of both the Indian monsoon system and the identification of supraregional

connections, SST reconstructions have been conducted in various regions of the Arabian Sea, as detailed in an overview by Gaye et al. (2018). Depending on the location, these reconstructions exhibited significant variations in SST (about 3 °C at maximum), responding differently to warm and cold periods, as well as local influencing factors such as atmospheric and oceanic circulations.

This study is centered around an alkenone-derived SST record (SL167) obtained from the northeastern continental

Oman margin in the Gulf of Oman in the northwestern Arabian Sea. This high-resolution sediment core spans the past 43 kyr, providing insights into the last glacial period and the transition to the current Holocene period. The new record is particularly significant since it represents the first high-resolution alkenone-derived SST record in the southeastern part of the Gulf of Oman. A compilation of SST records from the Arabian Sea by Gaye et al. (2018) showed that glacial SST were about 4 °C lower than during the Holocene. Further, they argue that the general glacial SST gradient within the Arabian Sea had a stronger

N-S insolation-driven component than during the Holocene with a more pronounced NW-SW circulation-driven component. However, records from the Arabian Sea show differences in their amplitude of SST changes and SST studies from the Gulf of Oman are still completely missing. Thus, this work aims to close the gap of past SST information in the Gulf of Oman and generally complete the past SST pattern in the Arabian Sea. Previous high-resolution SST records either do not extend as far into the past (e.g., Böll et al., 2015; Huguet et al., 2006), or exhibit lower SST resolution in the Arabian Sea (e.g., Schulte and

Müller, 2001). By comparing our SST data with existing records from the Pakistan Margin (93KL; Böll et al., 2015) (136KL; Schulte and Müller, 2001), Oman upwelling (MD00-2354; Böll et al., 2015) and the Horn of Africa within the Gulf of Aden (P178-15; Tierney et al., 2016), we aim to unravel the relationship between the SW and NE monsoons within the complex Indian monsoon system. Investigating the regional SST influences in the Gulf of Oman over the past 43 kyr is crucial for advancing our comprehension of regional climatic dynamics. Additionally, we compare the obtained SST data with

supraregional $\delta^{18}$O data from the eastern Mediterranean region (Sofular Cave; Held et al., 2024) and northern high latitudes (NGRIP; Svensson et al., 2008). This comparative analysis aims to elucidate global climate impacts in the Gulf of Oman during both, warm and cold periods.

## 2 Modern climate dynamics

During the Indian summer monsoon (SW monsoon), the heating of the Asian continent (low-pressure cell) and

development of a high-pressure cell over the southern Indian Ocean, lead to the development of strong, warm and moist low-level winds from SW direction.This drives surface ocean currents in clockwise circulation (Somali Current and East Arabian Current) in the Arabian Sea. In contrast, with the onset of the Indian winter monsoon (NE monsoon) the pressure gradient reverses due to stronger cooling of the Tibetan Plateau (high-pressure cell) than the warmer Indian Ocean (low-pressure cell). This results in moderate, dry NE winds and a switch to an anticlockwise surface ocean circulation (Findlater, 1969 Bansod et

al., 2003; Clemens et al., 1991; Clemens and Prell, 2003; Clift and Plumb, 2008; Findlater, 1969; Fleitmann et al., 2007; Schott et al., 2002; Webster et al., 1998; Webster, 2020; Wyrtki, 1973). SW monsoonal winds in spring and summer and the development of a clockwise circulation pattern induce seasonal upwelling of cold, saline and nutrient-rich deep water masses through Ekman transport, especially alongshore the eastern coasts off Oman and Somalia (de Boyer Montégut et al., 2007; Honjo et al., 1999; Izumo et al., 2008; Rixen et al., 2000). Ekman pumping lowers the SSTs in boreal summer by about 2-3

°C on annual average, compared to the northern, eastern and southern Arabian Sea (Levitus and Boyer, 1994). Almost simultaneously to SW monsoon conditions, NW winds transport dust plumes predominantly from the Arabian Peninsula as well as Iran into the Arabian Sea and can also affect the regional SST pattern, depending on their intensity and variability (Leuschner and Sirocko, 2000; Sirocko and Sarnthein, 1989). A substantial SST gradient of 4 – 5 °C develops over several hundred kilometers with the onset of the SW monsoon, creating a temperature low near the coast of Oman and a high of

approximately 29 °C in the western Gulf of Oman (Figure 2c). The northern, NW and NE Arabian Sea (north of 20 °N) indicate a clear seasonal SST signal with warmer temperatures during Northern Hemisphere summer, which rapidly decreases in fall and displays the lowest SSTs (ca. 23.5 °C) in the main stage of winter (Figure 2a; Dahl and Oppo, 2006; Kumar and Prasad, 1996; Levitus and Boyer, 1994).

The Arabian Sea High Salinity Water, Indian Ocean Central Water, Persian Gulf Water (PGW) and Red Sea Water

(RSW) constitute the four major sources of water masses in the Arabian Sea (Shetye et al., 1994). Dry air from the Himalaya during the NE monsoon causes high evaporative cooling, increases the density of surface water, and forms the Arabian Sea

High Salinity Water, particularly in the northern Arabian Sea (Kumar and Prasad, 1996, 1999; Madhupratap et al., 1996; Prasad and Ikeda, 2002; Shetye et al., 1994). The Indian Ocean Central Water, a combination of Indonesian Intermediate Water and Antarctic Intermediate Water, flows into the Arabian Sea through the SW Somali current (500 – 1500 m water depths) and becomes increasingly oxygen-depleted on its way to the Arabian Sea (Emery and Meincke, 1986; Resplandy et al., 2012; You, 1998).

Postglacial sea-level rise started the flooding of the Persian Gulf at around 14 ka, resulting in the transport of relatively young, warm, less oxygenated and saline PGW through the Strait of Hormuz (25 to 70 m) into the Gulf of Oman and the Arabian Sea (Lambeck, 1996; Shetye et al., 1994). According to Beni et al. (2024) rapid transgression phases in the western Persian Gulf centered around 10.4 ka and 9.2 ka and are followed by stable marine conditions since 8.8 ka. High saline PGW sinks below less-saline waters and produces a salinity maximum at depths between 200 to 400 m (Bower and Furey, 2012; Pous et al., 2004; Prasad et al., 2001; Premchand et al., 1986; Shetye et al., 1994; Wyrtki, 1973). Similarly, RSW is a warm, less oxygenated and high-saline water mass with an intermediate salinity maximum (500 – 1000 m water depths) in the Arabian Sea after flowing through the Strait of Bab al Mandeb (ca. 150 m) and mixing with Gulf of Aden water masses (Bower et al., 2000; Pathak et al., 2021; Rochford, 1964; Wyrtki, 1973).

In the Arabian Sea, mesoscale eddies, which represent cyclonic and anticyclonic rotating water masses (Figure 3), contrary to the surrounding main currents, emerge as key players in the regulation of surface ocean circulation (Al Saafani et al., 2007; de Marez et al., 2019; Fischer et al., 2002; Trott et al., 2019). Their upwelling and downwelling capabilities significantly affect the stratification of the upper ocean layers through the transport and redistribution of oxygen, nutrients, salinity and heat-driven or thermohaline water flows. The resulting influences on vertical and horizontal heat transport alter the regional and annual SST patterns in the Arabian Sea (Bower and Furey, 2012; Carton et al., 2012; Trott et al., 2019; Vic et al., 2015; Yao and Johns, 2010). Consequently, eddies that predominantly carry warmer waters can result in rising SSTs, while colder eddies can lead to a SST decrease. However, eddy-driven circulations are variable, transient and continuously moving, depending on the location and seasonal climate variations (SW/NE monsoon), implying that local fluctuations in SSTs may be intense and potentially temporary (Dong et al., 2011; L'Hegaret et al., 2016; de Marez et al., 2019; Piontkovski et al., 2019; Trott et al., 2019).

## 3 Material and methods

The piston core SL167 (741 cm long) was collected in the northeastern part offshore the Oman margin in the Gulf of Oman in the northwestern Arabian Sea (22° 37.15′ N, 059° 41.49′ E; 774 m water depth) during RV METEOR cruise 74/1b in September 2007 (Bohrmann et al., 2010). The core includes the period between about 3 to 43 ka. The sediment core was partitioned into sediment sample slices containing 2 cm of sediment. For the alkenone analyses 219 freeze-dried and homogenized samples were used.

## 3.1 Age-depth model of SL167

The age model published by Burdanowitz et al. (2024b) is based on twenty-one AMS [14]C dates of surface-dwelling planktic foraminifera (Figure 4). The Bayesian model package BACON v.2.5.6 by (Blaauw and Christen, 2011) in R (v.4.3, R Core Team, 2023) was used to maintain the age-depth model. Within the model the Marine20 calibration curve and a deltaR of 93 ± 61 years were used. For deltaR the weighted mean of two regional marine reservoir corrections from Muscat (Southon et al., 2002) from the marine calibration database (Reimer and Reimer, 2001) was used. The age uncertainties ranges from ± 170 to

220 years during the Holocene up to about ± 770 years in the oldest part of the record (Burdanowitz et al., 2024b).

## 3.2 Alkenone analyses

Alkenones were measured at 2 cm of sediment for the upper 162 cm and 4 cm resolution below 162 cm by combining consecutive subsamples, due to lower organic content (<1.5 %). To obtain the total lipid extract (TLE) about 3 to 18 g of

sediment were extracted by a Dionex Accelerated Solvent Extractor (ASE 200) using dichloromethane (DCM) and methanol (MeOH) (ratio 9:1) as solvent as described in Burdanowitz et al. (2024b). A known amount of an internal standard was added to the samples. During the extraction process, temperature and pressure were kept constant at 100 °C and 1000 PSI for five minutes. This procedure was performed three times. Each ASE 200 running sequence (17 to 18 cells in total) included a blank (combusted sea sand), a standard (combusted sea sand and internal standard) and a known working sediment standard. The

TLEs were rotary evaporated until almost dryness. Asphaltene separation was carried out using sodium sulfate ($Na_2SO_4$) column chromatography for separation of the hexane-insoluble fraction. The hexane-soluble fraction was saponified with 500 µl of a 5 % potassium hydroxide solution (KOH) in MeOH and placed in the oven for 2h at 85 °C. N-hexane was added to the saponified fraction, vortexed, followed by extraction of the upper, non-mixing neutral fraction. Then the neutral fraction was separated by column chromatography into apolar, ketone (containing alkenones) and polar fractions utilizing deactivated

silica gel (5 % $H_2O$) and different solvents (DCM for ketone separation). All samples were completely dried over night after each preparation step.

Quantification of alkenones was carried out by using a Thermo Scientific Trace 1310 gas chromatograph (GC), which used $H_2$ as carrier gas (35 mL min$^{-1}$) and is equipped with PTV injector (temperature 50 °C ramped with 10 °C s$^{-1}$ to 325 °C, splitless mode) and Thermo Scientific TG 5MS column (30 m, 0.25 mm thickness, 0.25 µm film). The GC is coupled to a

flame ionization detector (FID). GC-FID was programmed to held temperature at 50 °C for 1 min, then heat to 230 °C (20 °C min$^{-1}$), to 260 (4.5 °C min$^{-1}$) and to 320 °C (1.5 °C min$^{-1}$) where the temperature is held for 15 minutes. Identification of $C_{37:2}$- and $C_{37:3}$-alkenones was performed by comparing peak retention times of the samples with an internal working sediment standard and was followed by quantification through integrating the peak areas of $C_{37}$-alkenones and the internal standard (14-heptacosanone).

For calculation of the alkenone-based unsaturation index for $C_{37}$-alkenones we used the calculation by Prahl et al. (1988):

$$U_{37}^{k\prime} = \frac{C_{37:2}}{C_{37:2}+C_{37:3}} \qquad (1)$$

The $U_{37}^{k\prime}$ ratios were converted to SSTs by using the regional surface calibration of the Indian Ocean (Sonzogni et al., 1997):

$$SST = \frac{U_{37}^{k\prime}-0.043}{0.033} \qquad (2)$$

At least duplicate measurement was performed for each sample. The analysis of the duplicate measurement indicates an average precision of 0.1 °C.

**3.3 Statistical analyses**

We carried out spectral and wavelet analyses in R (v.4.3, R Core Team, 2023) to identify periodicities in the reconstructed SST data set. We used the REDFIT function of the package dplR v.1.7.4 (Bunn, 2008, 2010; Bunn et al., 2022) for the spectral analysis of the reconstructed SST. It is based on the Fortran 90 REDFIT source code developed by Schulz and Mudelsee (2002). For the wavelet analyses we used the R package biwavelet v.0.20.21 (Gouhier et al., 2021) using the morlet

wavelet function and bias-corrected power spectrum, which is based on Torrence and Compo (1998). The original data set has a resolution of 40 to 800 years (mean: 181 ± 124 years, median: 168 years) with lowest resolution during the Last Glacial Maximum (LGM; Burdanowitz et al., 2024b). Prior to the wavelet analysis, we first interpolated the reconstructed SST data to an evenly spaced data set by using the package ncdf4.helpers v.0.3-6 (Bronough, 2021) and the approx. function. In detail, we used the highest resolution of the record by using the function "get.f.step.size()" resulting into a resolution of 40 years

between two measurements. This results into a total of 890 time steps of an even spaced data set of the SL167 record.

**4 Results**

**4.1 Alkenone-based SST record of SL167**

Based on our SST calculations using the $U_{37}^{k\prime}$ index, we observed a range of approx. 7 °C, ranging from 27.4 °C to 20.1 °C (Figure 5a). SSTs were relatively high (26.3 to 27.4 °C) during several periods, including the mid-Holocene and

periods that can be chronologically attributed to the Bølling-Allerød interstadial (B-A) and D-O interstadial 4 – 9, and 11. Periods of low SSTs (20.1 to 25 °C) comprise the late and early Holocene including the 4.2 and 8.2 ka BP events. During the Pleistocene, the periods of lower SST can be assigned to the Younger Dryas (YD), and Heinrich event 2, 3 and 4 (H2, H3, H4). SSTs around the H4 (37 to 39 ka) are low but exhibit pronounced fluctuations of three to four degrees. The SSTs of the LGM (18 to 23 ka) do not show significant cooling. They remain relatively warm (>25 °C), with a short SST drop to about

24.4 °C between 19 and 20 ka. A marked increase of about 2 °C occurred during the mid-Holocene around 7.4 ka. Spectral

and wavelet analyses show significant periodicities ($\chi2 > 95$ %), including a 7200-year cycle and shorter periodicities of about 525- to 401-years (Figure 7a, b).

## 5 Discussion

### 5.1 Seasonality of SST in the Arabian Sea

190   Several studies from the Arabian Sea have shown, that alkenone based SST reconstructions reflect, at least for the Holocene, an annual mean temperature signal (Böll et al., 2014; Doose-Rolinski et al., 2001; Sonzogni et al., 1997). Therefore, we assume that our reconstructed SST record reflects changes in annual mean SST.Figure 2a–d illustrates the seasonal SST pattern in the Arabian Sea, highlighting the complexity and fluctuations throughout a whole year. Specifically, during the winter months and the NE monsoon period, the northern Arabian Sea (93KL and 136 KL), including the Gulf of Oman

(SL167), cools significantly (Figure 2a) and gradually warms up in spring (Figure 2b) and during the SW monsoon (Figure 2c). With the onset of autumn the northern Arabian Sea gradually cools again (Figure 2d). The SST pattern also highlights significant temperature fluctuations off the coast of Oman (MD00-2354), with the lowest temperatures during upwelling within the SW monsoon period (Figure 2c), but also somewhat lower SST in the northern part of the Oman upwelling during the winter months (Figure 2a). Warmest SSTs in the Oman upwelling occur during the intermonsoon season in spring and autumn

(Figure 2b, d). The western part of the Gulf of Aden (P178-15) only slightly cools during the NE monsoon and otherwise shows consistently warm temperatures.

   Our core location (SL167) in the Gulf of Oman is sensitive to SST changes throughout the year due to the influence of multiple factors. These are the intensity of the NE and SW monsoons (Figure 1; Figure 2a, c), the impact of the upwelling system along the southeast coast of Oman (Figure 2c), the influence of mesoscale eddies and their vertical and horizontal

thermohaline water flow (Figure 3a, b), the input from various water sources, as well as the development of a pronounced SST gradient between the Oman upwelling area and the Gulf of Oman/northern Oman margin (Figure 2c). The movement of the SST gradient during the summer months from west-to-east and vice versa significantly impacts the local SST signal, given its spatial extent of only a few hundred kilometers (Figure 2c). In order to differentiate regional relationships, differences and anomalies of the Arabian climate and monsoon cycle, we compare our SST record with other alkenone-derived SST records

from different areas in the Arabian Sea (Figure 5a–e), as well as additional regional proxy-derived climate patterns (e.g., $\delta^{18}O$ isotope data in speleothems) from the monsoon area and adjacent regions.

   The overall high variations of our alkenone-based SST of up to 7 °C during the last 43 kyr (Figure 5a) can be attributed to several climatic phases and events, which will be discussed in the following.

## 5.2 Sea surface temperature changes in the Gulf of Oman during the late Pleistocene and Holocene

### 5.2.1 SST variation during Heinrich Events

The SL167 SST record indicates a sharp temperature decrease during the H4 cold event, experiencing the lowest SST during the past 43 kyr. Further, the reconstructed SST minimum in the NW Arabian Sea marks the lowest SST compared with available records from the northern and western Arabian Sea (Figure 5a–e). This temperature drop might be linked to abrupt monsoon changes with an intense NE monsoon and/or NW winds, which could cause lower SSTs at the core site within cold stages. Studies from the northwestern Arabian Sea (e.g., Sirocko and Lange, 1991; Sirocko et al., 1991) have linked increased dust loads during the last glaciation to an amplified impact of NW winds. Consequently, during the cold event, intensified NW winds may have reduced SSTs by displacing warmer surface waters and promoting vertical mixing. Simultaneously, the impact of the SW monsoon decreases due to variations in solar radiation, resulting in a southward migration of the Intertropical Convergence Zone (ITCZ) and weakened ISM (Clemens et al., 1991; Godad et al., 2022; Prell and Kutzbach, 1992; Prell and van Campo, 1986). Besides the solar insolation and the NE/SW monsoon correlation, mid-latitude westerly winds also modulate monsoon conditions. Cold events can lead to a southward shift (south of the Tibetan Plateau) and intensification of westerly winds, resulting in an intensified NE monsoon and a fast retreat of the SW monsoon (Fang et al., 1999).

The analysis of oxygenation levels in both the water column (Figure 5f) and bottom water (Figure 5g) from the same core reveals different impacts of the H4 event (Burdanowitz et al., 2024b). While the observed increase in water column oxygenation corresponds with the decline in SST, the changes in bottom water oxygenation are less marked. The simultaneous strong oxygenation of the upper water column and decreasing SST indicate a significant impact of changing atmospheric rather than oceanic currents on SST in the study region. In addition, our SST record (Figure 5a) reveals a notable and highly fluctuating signal during and in the immediate aftermath of the H4 cold event. During this period, the SST signal is subject to a growing influence of the NE monsoon. Further, enhanced SW monsoon conditions can also strongly impact the SST signal, occasionally leading to substantial fluctuations. The oxygen minimum zone (OMZ) is more developed during the H4 event compared to its lower intensity during H3, H2 and H1, when the OMZ was comparably weak (Figure 5a, f). Note, that Allard et al. (2021), which we use as reference for the timing of Heinrich events, date the onset of H3 slightly earlier (32.7 and 31.3 ka) than the Greenland Ice Core records. The INTIMATE chronology (Rasmussen et al., 2014), using the nomenclature of Greenland Stadials (GS), related the GS 5.1 (around 30.6 ka) to H3 (Pedro et al., 2022), whereas GS 5.2 (around 32.0 ka) is falling in the timing of H3 by Allard et al. (2021).

At the beginning of the H4 event, the OMZ in the water column was even stronger than during the D-O Interstadials (see below) and the entire Holocene while the reconstructed oxygen levels in bottom waters (Figure 5g) was similar to the D-O Interstadials. It is conceivable that during the prolonged cold phase of H4 productivity was enhanced at the core site and led to the intensification of the OMZ.

### 5.2.2 Dansgaard-Oeschger Interstadials

Compared to the strikingly cold H4 event, higher SST characterize periods of moderate warming, during several D-O interstadials, including the D-O 11, D-O 4 – D-O 9 and B-A in the NW Arabian Sea. The warm interstadials typically exhibit only a short-term increase in SST (e.g., B-A, D-O 4) and are significantly more pronounced in the northwestern Arabian Sea compared to other Arabian Sea records (Figure 5a–e). No obvious warming trend was observed during D-O 2 and 3. During the warmer interstadials, the SW monsoon intensified while the NE monsoon weakened (Clemens et al., 1991; Prell and van Campo, 1986; Prell and Kutzbach, 1992), along with a northward shift of the ITCZ, due to a northward atmospheric energy transport across the equator (Schneider et al., 2014). The south-to-north movement of the ITCZ during D-O interstadials is associated with an increase in solar radiation and precipitation, indicating an opposite pattern to cold events (Cheng et al., 2012; Jaglan et al., 2021). In contrast, mid-latitude westerlies lose strength during interstadials and shift northward or remain entirely north of the Tibetan Plateau (Fang et al., 1999). Furthermore, the presence of low $\delta^{18}O$ values in speleothem records from Mawmluh Cave in India indicates elevated precipitation rates and intensified SW monsoon activity during a wet phase at 33.5 and 32.5 ka (Dutt et al., 2015; Jaglan et al., 2021), occurring almost simultaneously with the D-O 6 interstadial. Recent observations suggest a direct correlation between precipitation and temperatures, suggesting increased rainfall during warm interstadials and decreased precipitation during cold stadials (Allan and Soden, 2008; Trenberth et al., 2003). Despite the short duration of the D-O warming spikes, most of them show distinctly lower oxygen concentrations in the water column and bottom water (Figure 5f, g; Burdanowitz et al., 2024b).

Notably, reconstructed strong OMZ during D-O 10 and 4 events at the core site (Figure 5f; Burdanowitz et al., 2024b) are in line with somewhat lower SST. We attribute this to an enhanced influence of the SW monsoon winds and/or more northward-extended influence of the Oman upwelling area at the core site. Shortly after D-O 2 and with the onset of the LGM, at around 23 ka, an increase in SST is observed, which could be associated with D-O 2, similar to the findings at site 93KL (Figure 5d) and also supported by Böll et al. (2015). However, our record exhibits a distinct cold signal during D-O 2, which is even lower comparable to H2, and a subsequent SST increase. Burdanowitz et al. (2024b) noted a less pronounced OMZ in the water column but sub-/anoxic conditions in the bottom water at the core site. They attributed this to an intensified inflow of oxygen depleted RSW at intermediate depths and/or weak inflow of Antarctic Intermediate Water into the Gulf of Oman. However, stronger winds (NW/NE winds) could have facilitated enhanced mixing and ventilation of the water column, potentially contributing to the observed ventilation differences. Note that the age uncertainty during D-O 2 is about ± 350 years (Burdanowitz et al., 2024b).

### 5.2.3 Unusual SST pattern during the Last Glacial Maximum

Compared to other records in the Arabian Sea, the most unusual SST pattern at the core site occurred during the LGM where SSTs do not indicate a strong cooling, except for a minimum of about 24 °C around 19 ka (Figure 5a). The northern Arabian Sea (site 93KL; site 136KL) and the upwelling area (site MD00-2354) experienced a rapid SST drop. In contrast, the

NW Arabian Sea displayed a much lower decrease in SST compared to 93KL, and SSTs were not as low as observed at 93KL, 136KL, and MD00-2354 over the entire LGM period. Further, the water column was well ventilated during the LGM at the

core site, indicated by low $\delta^{15}N$ values (Figure 5f; Burdanowitz et al., 2024b). This phenomenon can possibly be attributed to intensified NE monsoon and NW winds as well as weaker SW monsoon, observed in large parts of the Arabian Sea during the LGM (Burdanowitz et al., 2024b; Duplessy, 1982; Jaglan et al., 2021; Sirocko et al., 2000). The lower glacial land temperatures in Central Asia (Annan and Hargreaves, 2013) and low boreal summer insolation (Böll et al., 2014, 2015; Gaye et al., 2018) resulted in an intensification of the NE monsoon and associated low SSTs. Previous studies have also sustained the hypothesis

of a weakened SW monsoon during the entire glacial period (Böll et al., 2015; Naidu and Malmgren, 2005; Schulte and Müller, 2001) and may offer a potential explanation for the moderately warm SSTs observed during the entire LGM. While a prolonged winter monsoon is anticipated for the LGM, the onset and related SST reduction in the NE Arabian Sea may be postponed due to its geographical location in relation to the northern Arabian Sea. This may account for the regional annual average SST contrast between the Pakistan margin (93KL, 136KL) and the Gulf of Oman (Figure 2). Further, elevated dust levels can also

lead to a decrease in SST at the surface (Yue et al., 2011). However, as stronger winds and input of dust could lower the SST, other factors may responsible for the moderate warm SST at the SL167 core site. In contrast, the SST was probably also influenced by mesoscale eddies, which could have controlled the transport of warmer waters into the region during the main period of the LGM. This eddy-driven influx likely resulted in elevated annual mean SST compared to other locations in the Arabian Sea. Furthermore, a pronounced eastward shift in the SST gradient during this period could have also influenced the

SST signal.

       Lowest LGM-SSTs between 19 and 20 ka at the core site are in line with other Arabian Sea records (P178-15, MD00-2354, 93KL; Figure 5b, c, d). However, SL167 (Figure 5a) show a faster increase in SSTs right after its minimum compared to the Oman margin and northern Arabian Sea, but subsequently aligns closely with the SST pattern from site P178-15P (Figure 5b). Both temperature records display a continuous rise in SSTs, at least until the midpoint of the B-A interstadial (~14 ka).

However, the northern Arabian Sea (93KL, 136KL) and the Oman upwelling region (MD00-2354) show a significant shift in warming, with the rise in SSTs beginning  around ~16/17 ka. Previous studies indicate that the intensification of the ISM and weakening of the NE monsoon at the end of the LGM led to a transition from a dry phase to a wet phase during the B-A interstadial (Böll et al., 2015; Dutt et al., 2015; Herzschuh, 2006; Jaglan et al., 2021). Warming of the high latitudes and the resulting reduction of the snow cover on the Tibetan Plateau is considered to be the most dominant factor (Herzschuh, 2006;

Overpeck et al., 1996; Wang et al., 2001; Zhou et al., 1999). NW winds, peaking between 15 and 13 ka (Sirocko et al., 2000), may have contributed to the earlier warming, even though SW monsoon conditions weakened (Leuschner and Sirocko, 2000; Sirocko et al., 2000). While the B-A interstadial indicates a strengthening of the SW monsoon, coastal parallel SW winds are too weak to produce upwelling, explaining the temperature increase observed in the upwelling area (MD00-2354) after the LGM (Böll et al., 2015; Huguet et al., 2006; Saher et al., 2007). Although IOCW and RSW intermediate and deep-water masses

may have had an impact on SSTs during this period, it is unlikely that their influence was substantial. This is because

coccolithophores are limited to the euphotic zone (0 – 150 m; Baumann et al., 1999, 2005) and IOCW and RSW occur at substantial depths.

Although SSTs in the western Arabian Sea continued the warming trend, a decline in SSTs was observed during the transition from the B-A interstadial to the YD period. Analysis of dust plumes in these regions reveals a marked reduction in dust input from the Persian Gulf, but only a minor decrease in the Central Arabia (Sirocko et al., 2b 000). Consequently, NW winds may continue transporting warm air masses to Central Arabia, whereas their impact on SSTs in the northern region declined. Furthermore, cooling of the Northern Hemisphere could also have played a vital role, resulting in the strengthening of the NE monsoon and weakening of the SW monsoon (Chen et al., 1997; Dutt et al., 2015; Fuchs and Buerkert, 2008; Herzschuh, 2006; Wang et al., 2001). The synchronous SST decrease in the northern (93KL, 136KL) and northwestern Arabian Sea (SL167) during this period, suggest a more substantial impact from the winter monsoon at the core site (SL167) during the YD compared to the LGM. This finding supports the hypothesis that the SST pattern may be influenced by variations in the intensity of NW winds, which can either strengthen or weaken over time. Moreover, it is worth noting that the inundation of the Persian Gulf, which began around 14 ka via the Strait of Hormuz (Lambeck, 1996) constitutes a crucial factor that must be taken into account, as it likely contributed to a significant decrease in SSTs during the YD period.

### 5.2.4 Strong and rapid SST changes during the Holocene

Fluctuations in SST are much more pronounced at site SL167 compared to all other regions of the Arabian Sea (Figure 5a–e). While other records predominantly exhibit glacial-interglacial cycles, our record stands out by high-amplitude millennial-scale SST oscillations. At the transition from the YD into the early Holocene, SSTs remained low (Figure 5a). With the onset of the early Holocene, SST at site SL167 increasingly resembled the SST signal from the Oman upwelling (MD00-2354). Similar SSTs from the Oman upwelling and the NW Arabian Sea are also observed during the early and late Holocene. The SST signal during the mid-Holocene exhibits a close correlation with the northern Arabian Sea cores (93KL, 136KL).

In contrast, during early Holocene, the SW monsoon intensified gradually in response to orbital forcing, i.e. intensification of summer insolation at 30°N with a maximum at around 11 ka. This was expressed in enhanced precipitation in Oman, Yemen, and south and southeast Asia (Dutt et al., 2015; Dykoski et al., 2005; Fleitmann et al., 2003, 2007; Fuchs and Buerkert, 2008; Herzschuh, 2006; Kessarkar et al., 2013). Low $\delta^{18}O$ values in speleothems indicate a rapid northward shift of the ITCZ and higher Northern Hemisphere temperatures, resulting in a stronger ISM and a weaker NE monsoon (Fleitmann et al., 2007).

The 8.2 ka cold event interrupted the warm and humid early Holocene period and weakened the ISM due to an amplified southward migration during this event of the generally northward-shifted ITCZ (Cheng et al., 2009; Dixit et al., 2014). Several studies suggested that invigorated SW monsoon winds led to a more vigorous upwelling during the early Holocene, which reflects lower SST and a $\delta^{15}N$ maximum (Böll et al., 2015; Rostek et al., 1997). These findings propose that strong SW winds move the water masses northward into the Gulf of Oman and affect the SST at the core site. Another study also suggested that these upwelled water masses were transported northward through gyres and eddies, affecting the oceanic

stratification in the Gulf of Oman (Watanabe et al., 2017). This is supported by the lower SSTs at site SL167 and MD00-2354 during the early and late Holocene. In response to a decrease in solar radiation, the ITCZ migrated continuously southward during the mid to late Holocene, accompanied by a continuous decrease in SW monsoon intensity and precipitation (Fleitmann et al., 2003, 2009; Fuchs and Buerkert, 2008; Gupta et al., 2005). The hypothesis of a stronger SW monsoon at the beginning of the early Holocene is supported by a recent study from the Persian Gulf (Beni et al., 2024) and evidence of increased wind activity between 9 and 6 ka during the boreal summer (Bassinot et al., 2011). Consequently, it is plausible that during the peak of the SW monsoon at the onset of the Holocene, increased water masses were transported from the upwelling region into the Gulf of Oman (Watanabe et al., 2017) and significantly lowered SST at SL167. However, lowest SST during the early Holocene occurred between 8.5 and 8.9 ka at the core site with a slight increase afterwards. Although the age uncertainty of our age model is around ± 220 years at this time, we are convinced that the age deviation from the 8.2 ka event is not due to age uncertainty. Evidence for this is another very well dated and high-resolution alkenone based SST record from the NE Arabian Sea (SO90-63KA) with similar SST pattern around that time (Burdanowitz et al., 2021).

With the beginning of the mid-Holocene, strengthening of NE monsoon conditions likely led to a temporarily interrupted transport of upwelled water masses to the core location. However, the increasing influence of NE monsoon conditions cannot be the sole driver of the rapid increase in temperature from about two degrees at 7.5 ka. One potential explanation for the observed changes in SST could be the inflow of water masses from the Persian Gulf into the Gulf of Oman. This hypothesis is supported by the fact that the Persian Gulf experienced increased flooding during this period and reached its present coastline at around 6 ka (Lambeck, 1996). However, the impact of PGW on the SST pattern in the Gulf of Oman may have been relatively small, given that the high-salinity PGW does not mix with the overlaying surface water, where the coccolithophores live (Baumann et al., 1999, 2005; Wyrtki, 1973). The strong SST gradient seems more likely to be the reason for the SST jump. An abrupt shift from a west-to-east SST gradient at approx. 7.5 ka may have increased the SST signal, followed by a gradual movement back in a westerly direction, resulting in a slow decrease in surface temperature.

The upwelling region exerted an increased influence from 5 ka onwards, evident from similar SST signals in the Gulf of Oman (SL167) and Oman upwelling area (MD00-2354) and suggest that an increased influx of upwelling water masses could have gradually shifted the SST gradient back in western direction. Beni et al. (2024) also argue that the NE monsoon became dominant after 6 ka. Consequently, it is plausible that the interplay between weakening SW winds (Bassinot et al., 2011) and a gradually strengthening NE monsoon abruptly prevented the transport of cold water masses from the upwelling region into the Gulf of Oman, preventing a further decrease in SST.

The decline in SST by about 5 ka coincides also closely with the transition to the end of the mid-Holocene climate optimum period, where the end of the African Humid Period falls within (Dallmeyer et al., 2013; Herzschuh, 2006), suggesting a possible connection between these phenomena. However, the Sahara did not dry out rapidly and consistently. While abrupt changes are observed in some areas (Demenocal et al., 2000; Tierney and deMenocal, 2013), other regions experienced a gradual, stepwise drying, emphasizing the non-linearity of the hydroclimate (Dallmeyer et al., 2020; Tierney et al., 2017). Similarly, Fleitmann et al. (2007) excluded an abrupt weakening of monsoon precipitation during the mid-Holocene in the

monsoon region. While a definitive connection cannot be proven, it also cannot be ruled out. However, it is more likely that the before mentioned processes played a more significant and influential role in shaping the observed changes.

During the mid to late Holocene transition period, the SST record captures a pronounced 4.2 ka BP event. Despite the prevailing aridity in significant parts of western Asia during this period (Giesche et al., 2019), SSTs demonstrate substantial variability throughout the transition and late Holocene. Consequently, as the SSTs fluctuate considerably, it becomes challenging to establish a conclusive link between a SST drop and the 4.2 ka event.

Overall, even during the Holocene, it becomes evident that we observe strong SST variations, with the cold events of
4.2 ka and 8.5 to 8.7 ka standing out as particularly significant signals with minor uncertainties surrounded by multiple SST fluctuations. This phenomenon could be attributed to several potential factors. The unique geographical and topographic features of the core site location in the Gulf of Oman may render it more sensitive to atmospheric and oceanographic changes, including pronounced local oceanographic processes such as currents and upwelling. Additionally, the region's geographic location and exposure could amplify the impacts of weather events, such as storms or strong winds. The coastal features and
topography of the Gulf of Oman could also contribute to faster warming or cooling of the water, particularly in shallower areas or near landmasses. Finally, alterations in ocean circulation patterns specific to the Gulf of Oman may result in increased SST variability by affecting the distribution of warm and cold water.

**5.3 Potential global drivers of SST variations in the Gulf of Oman**

The SL167 SST record exhibits periodic fluctuations consistent with millennial-scale oscillations, yet diverges from
glacial and interglacial changes. Notably, it demonstrates remarkable similarities extending beyond the Arabian Sea, as evidenced by ice core data from Greenland (Figure 6a; Svensson et al., 2008), and $\delta^{13}C$ time series of cave carbonates from the Mediterranean region (Figure 6b; Held et al., 2024). Most periods of lower SSTs (e.g., during YD and Heinrich events) in the NW Arabian Sea correlated with enhanced $\delta^{18}O$ values from NGRIP, indicating cold air temperatures in Greenland and the northern North Atlantic. Conversely, higher SSTs (e.g., during D-O interstadials) correlated with several lower $\delta^{18}O$ values,
characterized by moderate interstadial events in the North Atlantic Ocean. Based on these findings it can be inferred that the area of our core location is not exclusively shaped by local factors, but rather responsive to global temperature fluctuations that affect the SST signal.

Several studies have already suggested a close linkage between the North Atlantic Ocean, the AMOC and the Indian monsoon climate in the Arabian Sea based on the climate variability of the D-O cycles and Heinrich events (Leuschner and
Sirocko, 2000; Reichart et al., 1998; Schulte et al., 1999; Schulz et al., 1998; Sirocko et al., 1996). A weakening (strengthening) of the AMOC e.g., around the 8.2 ka cold event (or reverse: D-O interstadials) caused a southward (northward) shift of the ITCZ, which also suggests a decrease (increase) of the ISM (Cheng et al., 2009; Deplazes et al., 2014; Krebs and Timmermann, 2007; Zhang and Delworth, 2005). Although the monsoon strength is obviously linked to the North Atlantic and occasionally responds vigorously to abrupt climatic events (e.g., H4), the SST record of SL167 does not reflect all warm or cold periods
(e.g., no prominent cooling during the LGM). These findings demonstrate that NH cooling may influence the strength of the

SW/NE monsoon and SSTs, but other oceanic and atmospheric factors (mesoscale eddies, strong SST gradient and NW winds) can also have a crucial impact.

To identify any cyclic patterns in Gulf of Oman SST record and gain insights into the influencing factors, we conducted spectral (Figure 7a) and wavelet analyses of SST data (Figure 7b). The spectral analysis revealed significant periodicities of 7200 years ($\chi2 > 95$ %) in the SST data. This period could potentially be attributed to Heinrich events, which are characterized by large-scale melting of the Laurentide ice sheet and abrupt climate changes occurring over approximately 6.1 ka (Mayewski et al., 1997) and 7.0 ka (Calov et al., 2002). The alignment of our SST data with these periods obtained through spectral analysis supports this hypothesis. However, the wavelet analysis indicates that the prevalence of this period is not entirely evident, particularly during the interval from approx. 11 – 19 ka. Instead, this periodicity could potentially be attributed to oscillations in atmospheric [14]C, as suggested by Southon (2002). Their findings provide a rationale for the occurrence of archaeomagnetic coincidences within a 7 ka cycle, which is influenced by fluctuations in geomagnetic shielding as modulated by [14]C data. The 7200-year cycle may also be a subharmonic of the precession cycle, which was also suggested by Naidu et al. (2019).

Additionally, the spectral analysis revealed the presence of cycles of 7550, 4950-, 3750-, 2200-, and 950 years BP within the 90 % confidence interval ($\chi2 > 90$ %) as well. The presence of a 2200-year periodicity in monsoons was first observed in sediment records obtained off Oman (Naidu and Malmgren, 1995). This periodicity was attributed to interactions between oceanic circulation, atmospheric carbon fluctuations (Naidu and Malmgren, 1995; Thamban et al., 2007) and solar activity, estimated by tree ring records (Lean, 2002; Struiver and Brazinuas, 1993; Thamban et al., 2007). Our wavelet analysis reveals the presence of this cycle within the time intervals of ca. 18 – 5 ka and 35 – 22 ka. The remarkable reduction in cyclicity between 22 and 18 kyr BP, reflects stable LGM conditions and is a characteristic feature of this record which requires further investigation in the future. At the periodicity of 950 years, a widespread cycle emerges, supported by stalagmites from Oman (Neff et al., 2001), lake sediments from Alaska (Sheng Hu et al., 2003), and [14]C tree-ring data from the NH (Lean, 2002). These findings strongly indicate a primary solar influence on this cycle (Lean, 2002; Neff et al., 2001; Sheng Hu et al., 2003; Thamban et al., 2007).

The results of the spectral analysis indicate the presence of periodicities of 525, 505, 493, 486, 471, 437, 409, and 401 years ($\chi2 > 95$ %) as well as 427, 420 ($\chi2 > 90$ %) in our dataset. These relatively short periodicities are predominantly present in the Holocene and have been observed and documented in global records. Thus, several studies, such as the analysis of [14]C tree rings (Struiver and Brazinuas, 1993), demonstrate that the 500-year periodicity is attributed to changes in ocean circulation, especially of Atlantic deep-water formation (Bhushan et al., 2001; Kessarkar et al., 2013). Considering the close relationship between Asian monsoons and the position of the ITCZ, the 500-year periodicity could be closely linked (Kessarkar et al., 2013). The other periodicities also appear to be associated with solar cycles (Menzel et al., 2014). Loutre et al. (1992) suggests that cycles of 432 years (88 % probability) correspond to eccentricity periodicities, thus at least the 437-year cycle in our dataset can be attributed to them. Although there are some differences in the other cycles (+/- 30 years), the possibility of a correlation should not be disregarded. However, it is important to note that the resolution of these numerous short periods is

not very precise (Figure 7b). This is particularly evident around and during the LGM (approx. 15 – 23 ka), where no clear periodicity can be identified. Such gaps may reflect chronological uncertainties and resolution limits in this part of the record. Further these short periods can potentially be assigned to the cyclicity peaks but could also reflect inherent uncertainties in the age model, which range from 170 – 240 years for the Holocene and up to 770 years for the oldest part of the record.

## 6 Conclusion

In this study, we present a high-resolution alkenone-based SST record from the Gulf of Oman spanning the past 43 kyr. The SST reveals significant temperature fluctuations of about 7 °C, reflecting diverse climatic influences, and demonstrating increased sensitivity to climate variations compared to other Arabian Sea core locations. Thus, we provide the first SST data from this unique region and further complete the SST pattern in the Arabian Sea. The most prominent cold phase occurred during the H4 event with SSTs down to about 21 °C. Further cooler SST phases are reconstructed during the H3 event, the period between 19 and 20 ka, YD as well as the 8.2 ka and 4.2 ka event. We attribute these SST declines to reduced solar radiation and a southward ITCZ shift from a weakened AMOC, leading to strengthened mid-latitude westerlies and NE monsoon conditions while weakening the SW monsoon. Conversely, SSTs remain warm during D-O 11, D-O 4 – 9 and B-A, marked by increased solar radiation and a northward ITCZ shift, intensifying the SW monsoon and weakening NE monsoon conditions.

One of the most striking feature of our record is the absence of a strong SST decrease during the LGM, which is markedly distinct from previous SST reconstructions in the Arabian Sea, where a more pronounced drop was consistently observed. This may be linked to a weakened SW monsoon and a reinforced NE monsoon. Yet, enhanced NW winds, warmer eddy currents, and an SST gradient shift in the Gulf of Oman, significantly influence SST during this cold period. Compared to other Arabian Sea SST records, our record reveals strong rapid SST fluctuations throughout the Holocene by about 4 °C. The 8.5 – 8.7 ka and 4.2 ka events are marked as cold SST events at the core location. Further, a strong rapid increase of SSTs of about 3.5 °C within about 1200 years during the mid-Holocene SST may be attributed to an abrupt eastern shift in the SST gradient.

The SL167 SST record highlights the sensitivity of the Gulf of Oman to climatic variations. Specifically, it emphasizes the complex interaction of monsoonal and oceanographic processes influencing SST variations due to the unique semi-enclosed location at the gateway of the Gulf of Oman and the Arabian Sea. The location of the Gulf of Oman makes it an important area for understanding the general climatic mechanisms in the region. Future research, particularly in the Gulf of Oman, is essential to better understand the complexity of this system. Additionally, placing these findings within a broader regional framework, which includes neighboring marine and atmospheric systems, will improve our understanding of past climate/oceanic dynamics. This approach will help to reveal how changes in the Gulf of Oman influenced and were influenced by larger climate patterns in the Indian Ocean and surrounding areas.

## Appendix

### Data availability

The alkenone based SST dataset is stored and available at PANGAEA under https://doi.org/10.1594/PANGAEA.967645 (Burdanowitz et al., 2024a).

### Author contributions

JM: conceptualization, formal analysis, investigation, methodology, visualization, writing − original draft preparation. NB: conceptualization, formal analysis, investigation, methodology, visualization, writing – original draft preparation. GS: conceptualization, resources, supervision, writing – original draft preparation. BG: conceptualization, supervision, writing – original draft preparation.

### Competing interests

The contact author has declared that none of the authors has any competing interests.

### Acknowledgements

This research is funded by the Deutsche Forschungsgemeinschaft (DFG, German Research Foundation) through Germany's Excellence Strategy – EXC 2037 'CLICCS - Climate, Climatic Change, and Society' – Project Number: 390683824, as part of
495 the contribution to the Center for Earth System Research and Sustainability (CEN) at Universität Hamburg. The monthly SST visualizations used in this paper were produced with the Giovanni online data system, developed and maintained by the NASA Goddard Earth Sciences Data and Information Services Center (GES DISC). We acknowledge and appreciate their contribution to the availability and accessibility of valuable data for our research. We convey our thanks to Ocean Data Lab for supplying the mesoscale eddies data utilized in this paper. We thank Hartmut Schulz for his support during core recovery and sampling.
We express our gratitude to Frauke Langenberg, Marc Metzke, Miriam Warning and Sabine Beckmann for providing technical

and analytical support. Further, we thank the three anonymous reviewers for their constructive and helpful comments which helped us to improve the paper.

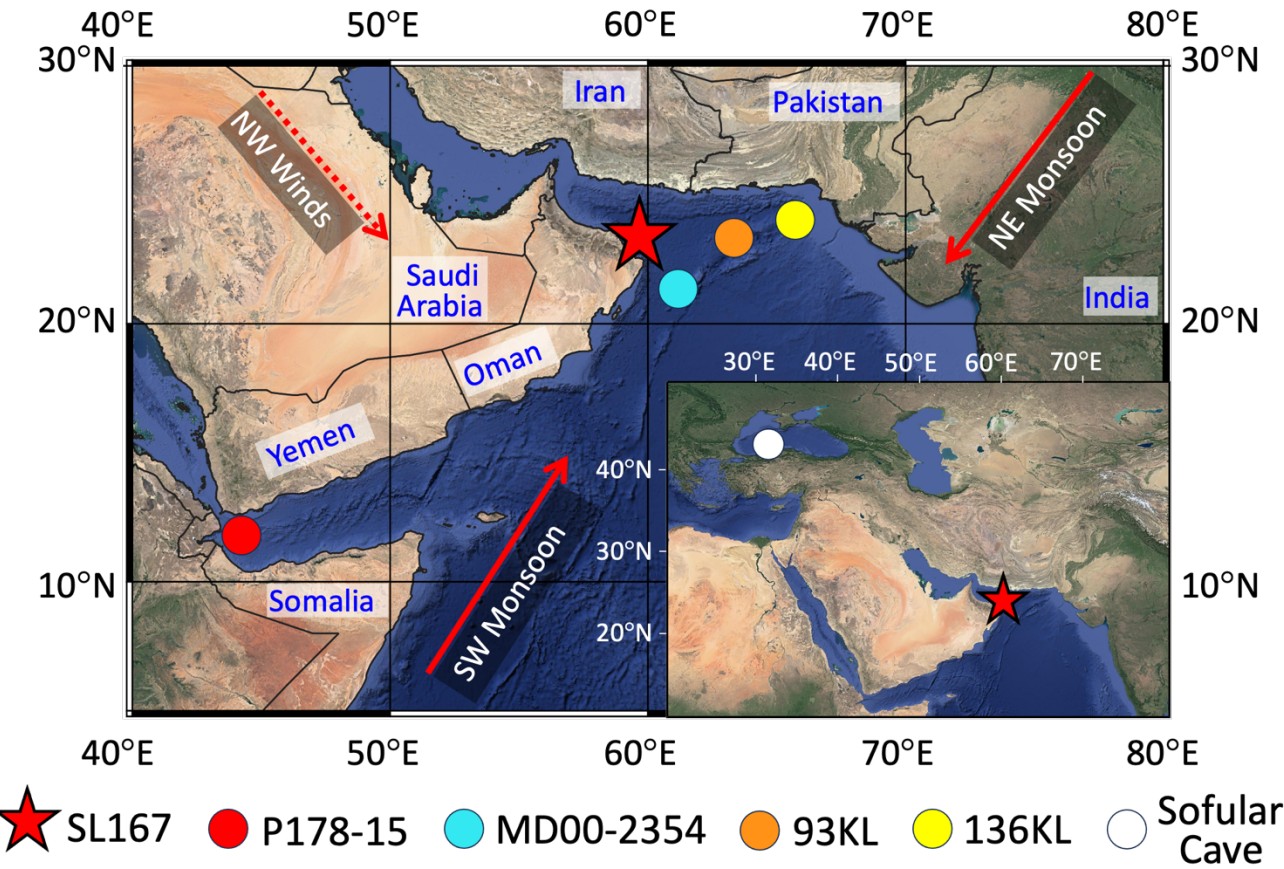

**Figure 1: Map of the Arabian Sea with the location of the study site SL167 (red star) from northwestern Arabian Sea offshore Oman, core P178-15 (red dot) from the western Arabian Sea** (Tierney et al., 2016)**, core MD00-2354 (blue dot) from the Oman upwelling** (Böll et al., 2015)**, core 93 KL (orange dot) from the northern Arabian Sea** (Böll et al., 2015)**, 136KL (yellow dot) from the northern Arabian Sea** (Schulte and Müller, 2001) **and stacked record from Sofular Cave (white dot in the inset map) from northern Turkey** (Held et al., 2024)**. Red arrows are used to represent the dominant wind pattern during southwest (SW) monsoon and northeast (NE) monsoon. Northwest (NW) winds are represented by the red dashed arrow. The map was created using QGIS v 3.28.3 from © Google Earth (wms layer (last modification February 2024): https://mt1.google.com/vt/lyrs=s&x={x}&v={y}&z={z}) and geoBoundaries (shapefile data (last modification February 2024): https://www.geoboundaries.org/globalDownloads.html ).**

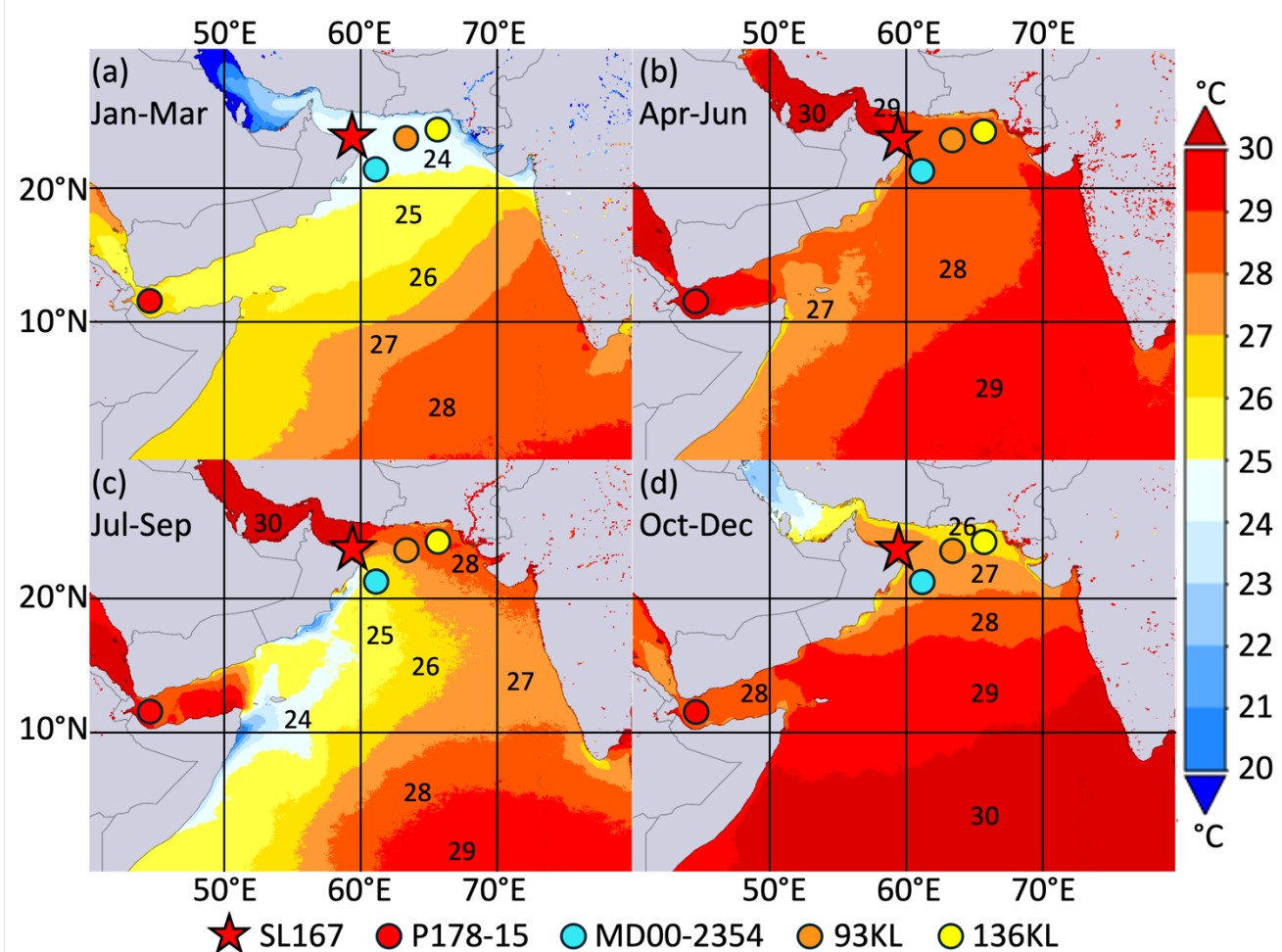

**Figure 2: Map showing monthly recurring averaged SST data from 2002 – 2023 during (a) the Indian winter monsoon season (IWM; January – March), (b) April and June, (c) the Indian summer monsoon season (ISM; July – September) and (d) October and December using [MODIS-Aqua MODISA_L3m_SST_Monthly_4km vR2019.0] satellite data from Giovanni v 4.38. Red star marks the location of study site SL167 in the northwestern Arabian Sea, offshore Oman. Red dot indicates core P178-15 from the western Arabian Sea** (Tierney et al., 2016). **Blue dot represents core MD00-2354 from the Oman upwelling region** (Böll et al., 2015). **Orange dot shows core 93 KL from the northern Arabian Sea** (Böll et al., 2015). **Yellow dot marks core 136 KL from the northern Arabian Sea** (Schulte and Müller, 2001).**

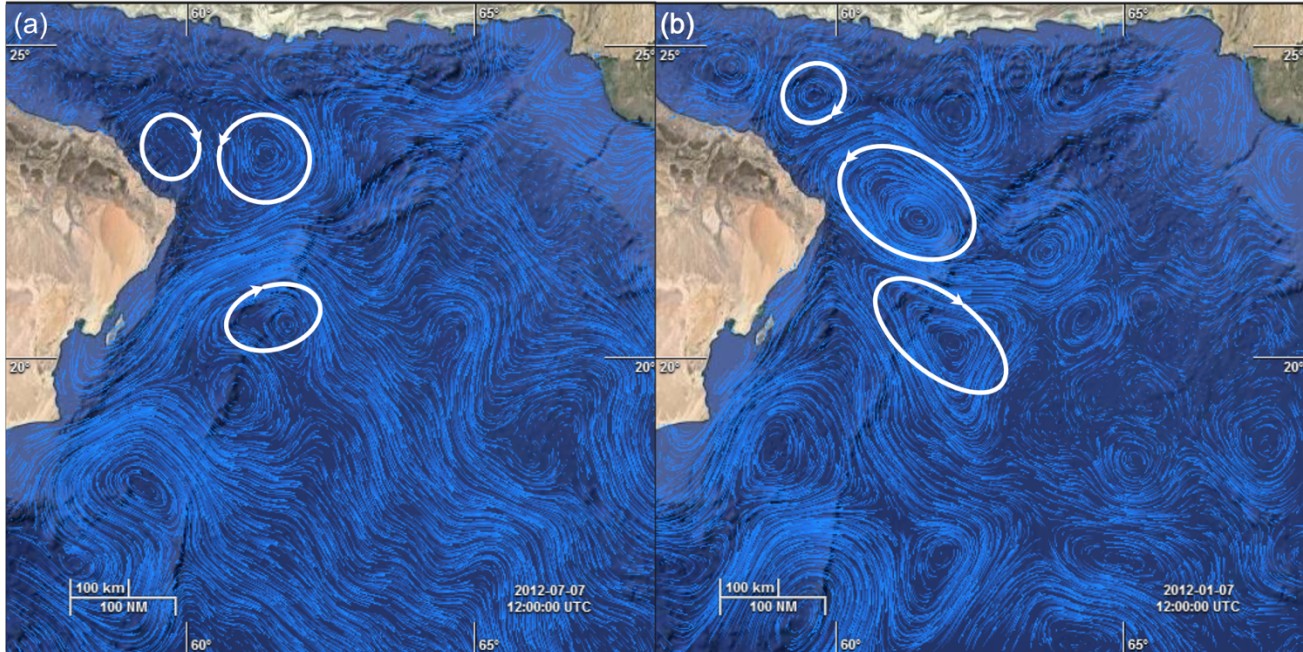

**Figure 3: Snapshot of mesoscale eddies during different seasons: (a) July 2012 during the SW monsoon and (b) January 2012 during the NE monsoon season. Figures were generated with Ocean Data Lab (https://ovl.oceandatalab.com/, accessed on 23.09.21) using the "total 15 m current streamline (Globecurrent, CMEMS)" product. The white circles with arrows denote the current streaming direction.**

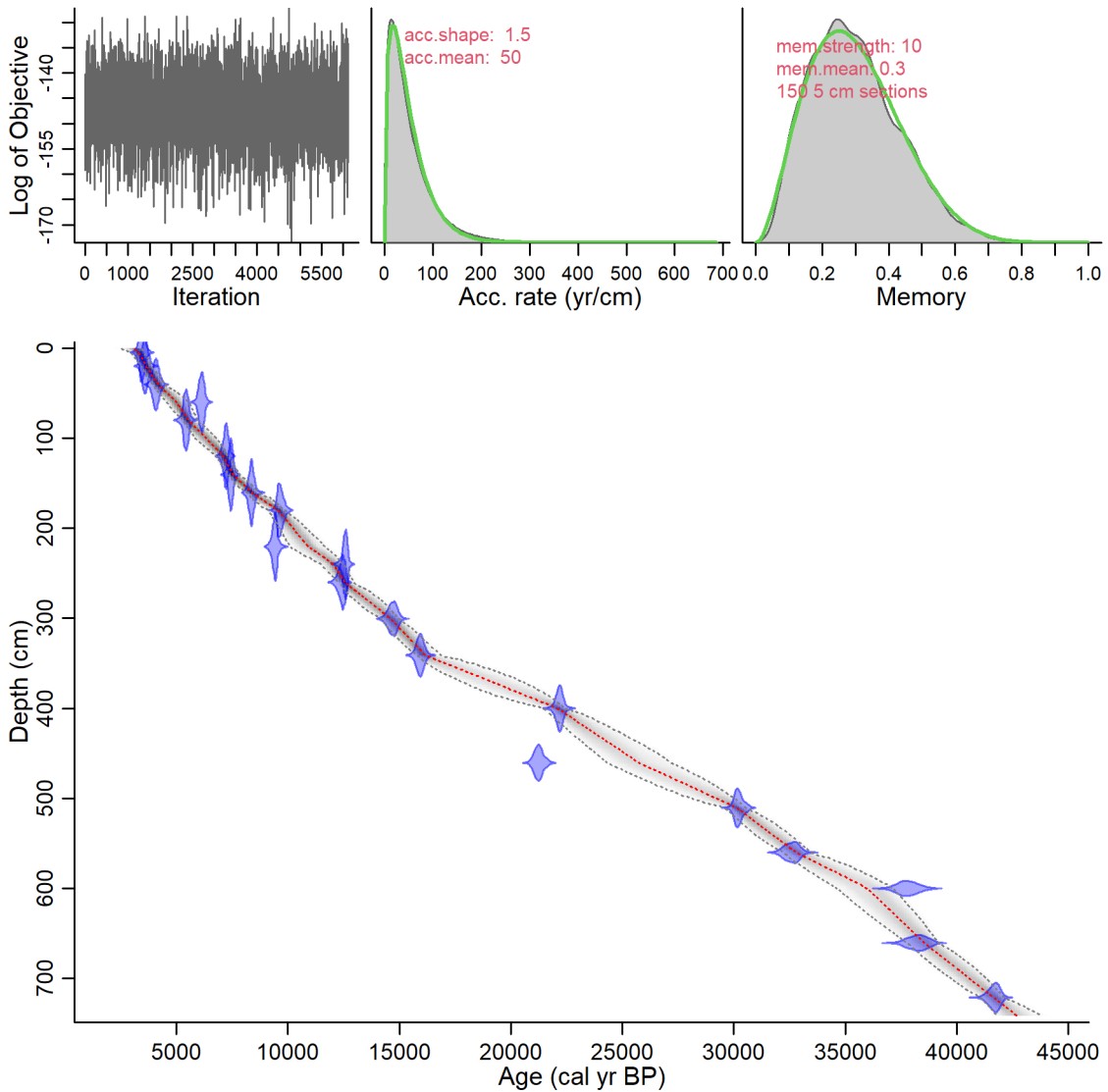

**Figure 4: Age-depth model of SL167 by** Burdanowitz et al. (2024b) **using the R package BACON v. 2.5.6** (Blaauw and Christen, 2011).
**The upper panels (a–c) show the Markov chain Monte Carlo iterations (a), the distributions of the prior (green curve) and posterior**
**(grey area) accumulation rates (b), and memory (c). The lower panel (d) shows the age-depth model of SL167. The calibrated ¹⁴C**
**dates are shown in blue. The red line shows the modelled mean age of SL167 with the 95 % confidence interval (dotted black lines).**
**A deltaR of 93±61 years was applied.**

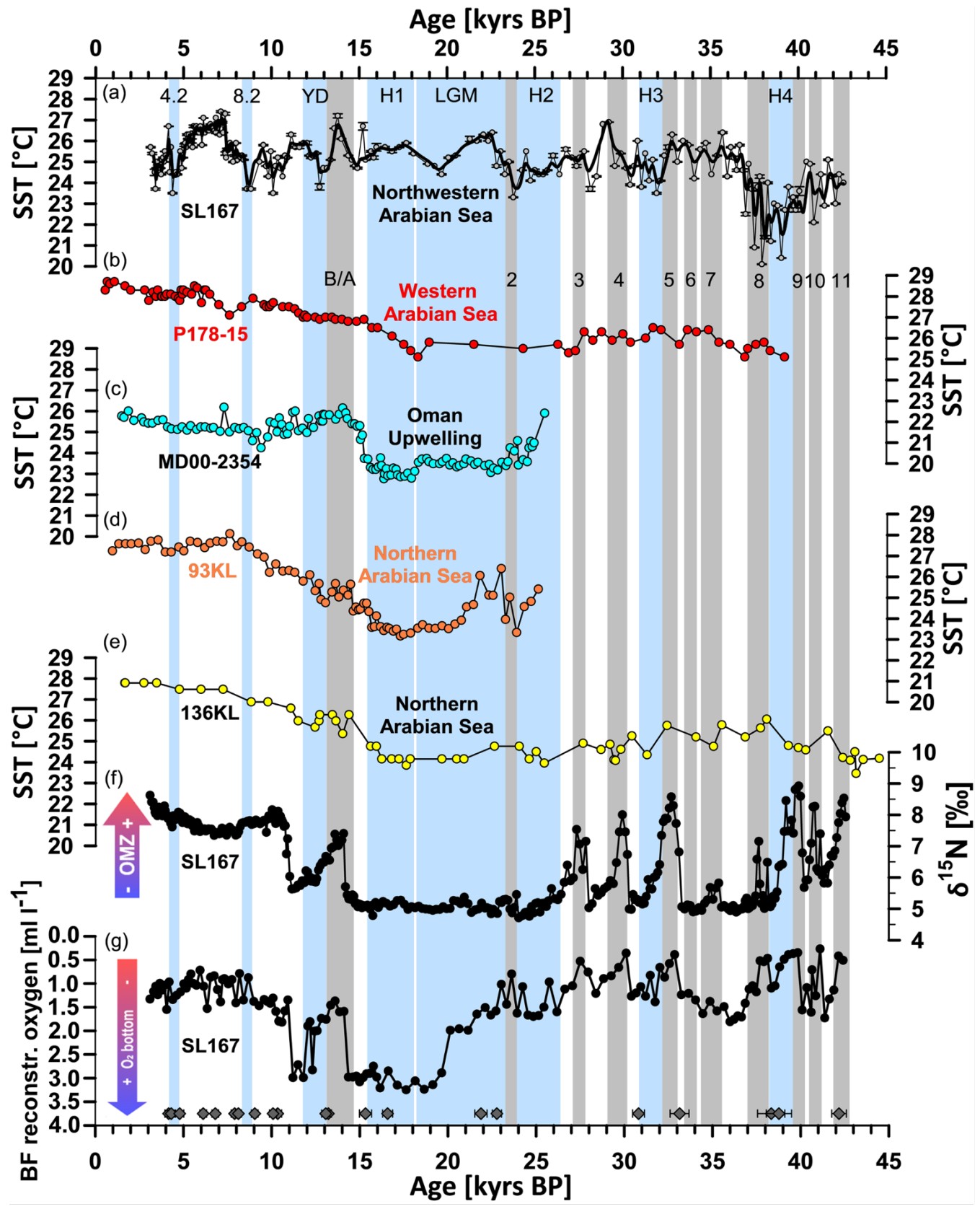

**Figure 5: Contrasting SST and monsoon records in the Arabian Sea over the last 45 kyr. a) SST of SL167 (this study) from northwestern Arabian Sea offshore Oman, b) core P178-15P** (Tierney et al., 2016) **from the western Arabian Sea, c) core MD00-2354** (Böll et al., 2015) **from the Oman upwelling, d) core 93KL** (Böll et al., 2015) **from the northern Arabian Sea and e) 136KL** (Schulte and Müller, 2001) **from the northern Arabian Sea. f) Nitrogen isotopes (δ$^{15}$N; SL167) serving as an indicator for denitrification and strength of the Oxygen Minimum Zone (OMZ) and g) the ratio of (lycopane + n-C35)/n-C31 (SL167) indicating bottom water oxygen levels (both published in** Burdanowitz et al. (2024b). **Blue bars indicate the 4.2k and 8.2k event, the Younger Dryas (YD), the Last Glacial Maximum (LGM) and the Heinrich 1 (H1), Heinrich 2 (H2), Heinrich 3 (H3) and Heinrich 4 (H4) event, Timing and duration of Heinrich stadials after** Allard et al. (2021)**; Grey bars indicate the Bølling-Allerød (B-A) interstadial and Dansgaard-Oeschger (D-O 1 – D-O 11) interstadials after** Fleitmann et al. (2009). **Diamonds exhibiting dated ages of SL167 after** Burdanowitz et al. (2024b).

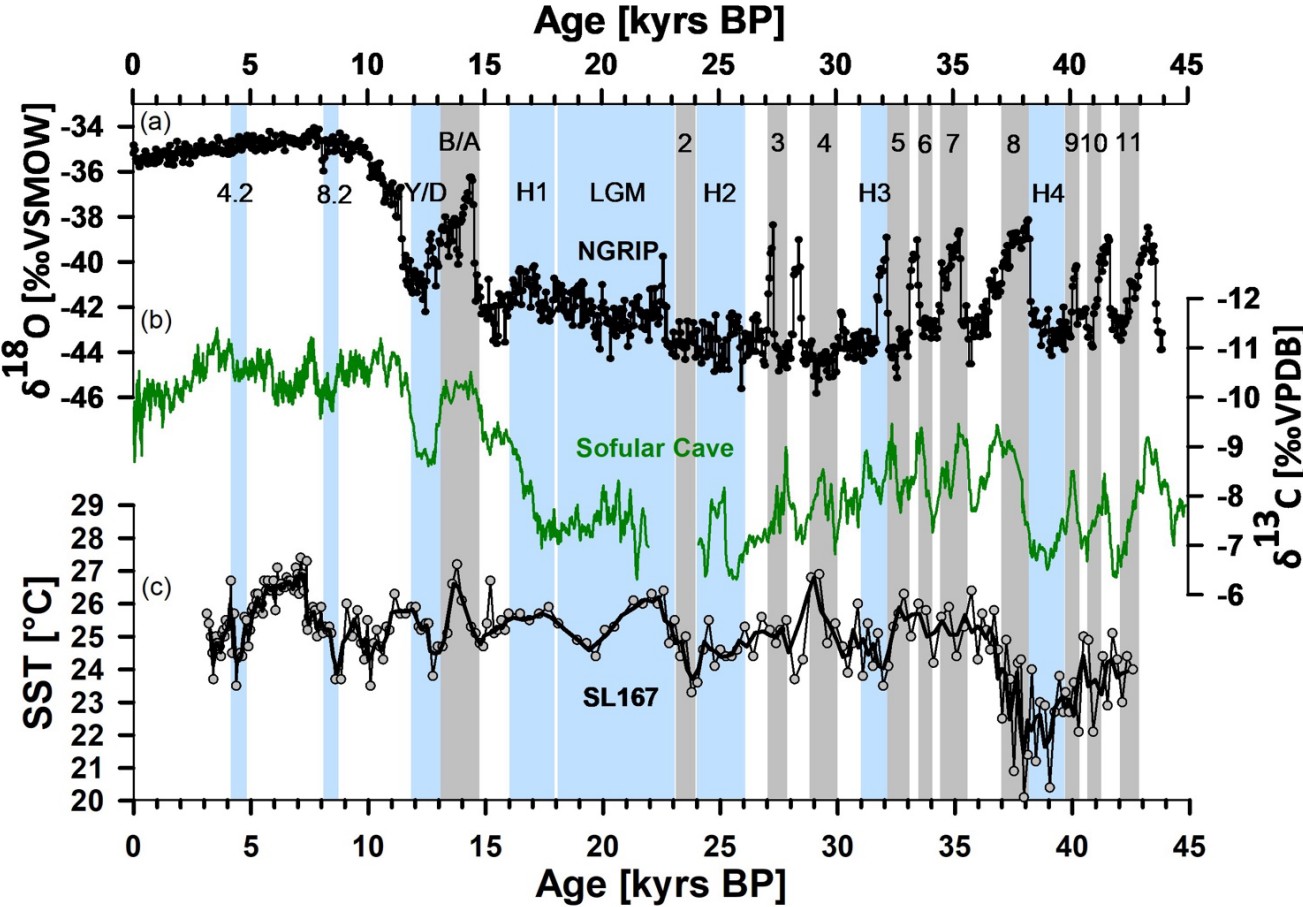

**Figure 6: (a) NGRIP δ$^{18}$O time series, derived from the northern Greenland ice core** (Svensson et al., 2008). **(b) δ$^{13}$C and time series of a stacked record from Sofular Cave in northwestern Turkey** (Held et al., 2024) **and (c) alkenone-derived SST of SL167 (this study) from northwestern Arabian Sea. Blue bars indicate the Younger Dryas (YD), Last Glacial Maximum (LGM) and the Heinrich 1 (H1), Heinrich 2 (H2), Heinrich 3 (H3) and Heinrich 4 (H4) event; Grey bars indicate the Bølling-Allerød (B-A) interstadial and Dansgaard-Oeschger (D-O 1 – D-O 11) interstadials after** Fleitmann et al. (2009).

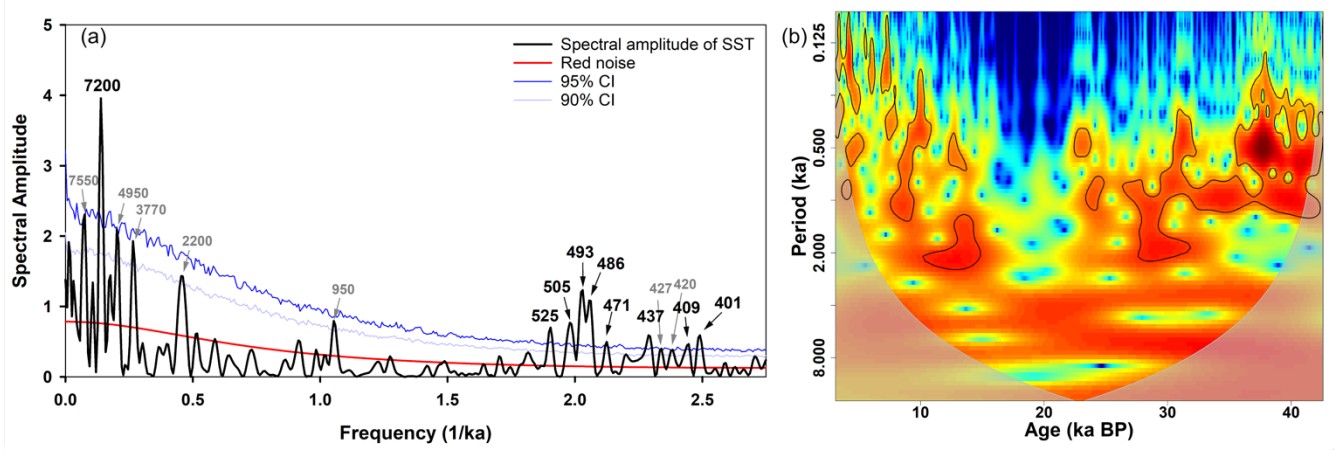

Figure 7: Spectral analyses (a) of SST amplitude of SL167 (Spectral amplitudes given in years and frequencies in 1/ka). The grey shaded area is showing the cone of influence and red (blue) colors represents high (low) power of the wavelet power spectrum. The black line denote the 95 % significance level. The Wavelet Analysis Visualization (b) depicts the time-frequency profile of a signal through wavelet transformation, featuring various signal characteristics distinguished by colors, offering insights into both the temporal and frequency-related aspects of the analyzed signals. The blue line represents the cone of influence, while the black lines denote the 95 % significance level.

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
