# Peer review of "Spatial and temporal variability of sea surface temperatures and monsoon dynamics in the northwestern Arabian Sea during the last 43 kyr"

_EGUsphere, 2024_

## Referee Comment (RC3)

The study by Maier et al., "Spatial and temporal variability of sea surface temperatures and monsoon dynamics in the northwestern Arabian Sea during the last 43kyr" would be a significant contribution based on Alkenone proxy in the Arabian Sea.

There are only few studies based on Alkenone proxy in the study area and the present study with high resolution, long SST record is interesting work in the complex region and is accepted subject to minor modifications. There are few suggestions which authors need to address prior its acceptance.

Major comments

Gulf of Oman, is a complex region and specifically influenced by SW monsoon due to its location and regional factors. Studying the region where seasonality is less pronounced due to the upwelling induced cooling during the summer season. Can the present study mark the variations in the seasonality of the two monsoon periods. Please elaborate on this aspect.

It is assumed that the alkenone proxy used in the study provides an Annual Mean SST. Does it not bias the study, can this give a true measure of the seasonality in the study area?

Annual SST variations are combination of different temperature signals such as Solar insolation and evaporative cooling, Strong NE winds, intensity of Upwelling, thus overcoming the bias by each signal would be difficult to estimate.

Past studies based on alkenones and other proxies from Arabian Sea reveals cooling during the LGM which varied regionally.

"The Unusual SST pattern at LGM" is not explained properly, majorly all the records in the Arabian Sea suggest cooling of atleast 2°C at LGM. There are several reasons as discussed in this section which prompts towards reduced SST contrary to what has been given as justification for warm SST at LGM, for example

    (i)     When there is an intensified NE monsoon effect the associated SST would be lowered

    (ii)    Reduced solar insolation compared to Holocene

    (iii)   Weakened SW monsoon during the entire glacial period cited references such as Boll et al., 2015; Naidu and Malmgren, 2005; Schulte and Müller, 2001, also suggest reduced SST at LGM.

    (iv)   Elevated dust levels could lower the SST then why there is a warm SST at the core site?

Infact, at the YD the low SST was explained with these reasons, compared to LGM's warm SST anomaly. All these points indicate towards lower SST's. Please provide proper justifications of warm SST at LGM.

Is there any time lag similar to what has been discussed by Naidu and Malmgren, 2005 or the chronology of the sediment core needs to tested for its robustness. Since in few studies the increase in sedimentation rate has been reported at LGM for various reasons in the basin.

The chronology part is discussed in Burdanowitz paper, however there should be an age model in the present study.

Minor Comments

Line no. 62, $\delta^{13}O$ needs to be corrected to "$\delta^{18}O$"

References should be in chronological order (for eg., line no. 72-75), this needs to be corrected throughout the manuscript. At several instances similar correction has to be made.

Line no. 90, "500 -1500m", add space between the no. number and units. Similar comment for the line 96, 98, 99, then at 120 and at several other. This need to be corrected throughout the manuscript.

The initial sampling was done at 2 cm interval. However, the alkenone analysis has been done only for 219 samples. The reason stated behind this the concentration of organic matter. However, is there any difference in terms of resolution of the study at later half of the core?

At line no. 199 "Enhanced SW monsoon conditions can also strongly impact the SST" If the SST is Annual average then how to distinguish the seasonality?

Why there isn't any trend in increase in SST at DO 2 and DO3?, Please provide justification for these two interstadials.

In general, for DO Interstadials strong NW monsoon was predicted, however the authors didn't mention about the effect of upwelling induced cooling on the SST in these time periods. Specifically looking at DO2 the low SST justification provided is due to intrusion of RSW or AIW in the Oman margin, with stronger mixing forced by NW/NE winds. However, at large at all the interstadials the SW monsoon winds were prevalent then what led to the change in contrasting wind conditions at DO2 interstadial? Please elaborate.

Is there any specific reason behind the increase in SST after DO2 and at the onset of LGM at 23 Ka.

At line no. 246, Moreover, during the LGM it has been postulated that there was an increased transport of warm water into GOM. The authors didn't provide any justification for the statement.

Global Factors influencing the SST variations at Gulf of Oman
In this section, the present dataset compared with records of other regions show similar variations. However, at LGM variation doesn't match with global records and the authors suggest influence if regional processes. What significant factors change this trend when comparing the records at other time periods such as Heinrich events, DO and BA cycles, early Holocene etc. which doesn't apply compared to LGM?

The spectral analysis periodicities of 7200 in the SST data which is attributed to Heinrich Events due to changes in Laurentide ice sheet. However, the authors may look at Naidu et al., 2019 wherein changes in the cyclicity is attributed to precision.

Conclusion

The significant points of the present study should be elaborated rather just mere stating the results of the study.

In general the dataset is very interesting but the discussion lacks proper reasoning and interpretation and the authors are encouraged to touch upon the comments to provide a better insights on their study.

---

## Author Comment (AC1)

**Comments Anonymous Referee #1:**

The article by Maier et al. presents a 43 ka-long alkenone-based SST record in the NE corner of Oman peninsula. The core location is hence potentially under the influence of the Mediterranean region, and the Asiatic and African landmasses, so many different climatic regimes could interfere, manifest, or be masked by a multitude of processes. Then it is not surprising, and very interesting, to see a SST record that look anything like what has been already published in the literature in the broader NW Indian ocean sector.

I find the authors have done a good job in interpreting their curious SST record, and I liked their choice of showcasing their record along with other alkenone-based records that are used in Figure 4 to appreciate the potential contrasted influences that surely played some role in shaping their SST record. I think the article could be published after minor revisions.

First, I suggest the authors to introduce more clearly the complexity of their SST record that comes from such a locality, and develop more on that in the conclusion. As it stands the conclusion only summarizes the main findings, but I think there is room to finish the conclusion within a broader regional context than the Arabian Sea and the Gulf of Oman only.

**Response:** First, we want to thank the anonymous referee for his/her helpful and thoughtful comments to improve the manuscript. To introduce the complexity of our SST more clearly we will add a further paragraph at the beginning of the discussion and the conclusion. We agree that this important aspect is indeed missing in the conclusion. Further, we will rearrange the conclusion and add an additional paragraph to bring our SST record into a broader regional context as suggested.

Other remarks:

What I found really missing was a stronger description of the age model, perhaps through adding an additional figure. Before starting to read the article the only d15N record looks so much like a series on Bond cycles for the MIS3 that I first thought that the Heinrich events were mislabeled. I had to check in the original Burdanowitz paper to be convinced by your age model, and I think it is really missing in your own article.

**Response:** We agree with the reviewer. To describe the age model of SL167, we will add the age-depth model based on AMS $^{14}$C dates of planktic foraminifera published in Burdanowitz et al. 2024, where we first described the age model. Further, we will add the measured ages as points with its uncertainties into Figure 4. However, as a detailed description is already available in Burdanowitz et al. 2024, we will only add the reference link to the age-depth data (Pangaea data base) and will not republish the table in the current manuscript.

On your SST interpretation, with which I have no problem, it is sometimes hard to follow when you describe your curious SST record with other ones in chapters 5.1.2, 5.1.3 and 5.1.4. Particularly, for example, when you discuss the seasonality of climate patterns such as the monsoon and ITCZ along with other SST records in the northern Arabian Sea. I think the authors will find guidance if you also show the monthly SST for all individual sites that you show in Figure 4, to better highlight how changes in other SST could be partly driven by

changes in seasonal/atmospheric processes that you describe. For example, the paper by Bassinot et al. (https://doi.org/10.5194/cp-7-815-2011) shows how seasonal changes in wind could enhance/dampen upwelling in regions situated in the western/eastern parts of the Arabian Sea. Even if you consider your own alkenone as being reflective of mean-annual SST I think more discussion on how seasonal features can deeply affect the regional dynamics could be more apparent in your discussion.

**Response:** The reviewer raised a justified concern about the seasonal variability of atmospheric patterns. First, we will extend Figure 2 showing now the mean SST for a) January-March, b) April-June, c) July-September and d) October-December. Second, we will add the individual sites from Figure 4 into Figure 2 to show the seasonal SST variability of each site. Third, we will discuss the differences and seasonal variability in more detail at the beginning of the discussion and include this critically in the further discussion.

The combined proxy-model study on the Holocene changes in the wind and productivity patterns in the Arabian Sea area by Bassinot et al. (2011) nicely underlines the hypothesis of stronger SW monsoon impact on our SST record during the early Holocene. They found stronger SW monsoon induced winds during the summer months at the Oman Margin inducing stronger upwelling at around 9 ka compared to 6 ka. It is plausible that strong upwelling and winds increase the water mass transport from the colder upwelling region into the Gulf of Oman (Watanabe et al., 2017) and significantly lowering SST at site SL167. Further, the decreasing strength of SW winds from 9 to 6 ka (Bassinot et al., 2011) could have reinforced the rapid SST increase at site SL167 as less colder water masses from the Oman upwelling arrived at the core site.

However, we are aware that the core site is strongly affected by the SW monsoon, the NE monsoon and also NW winds. Strong changes of each of them may bias the, reconstructed annual SST towards slightly more seasonal SST.

I liked the proposition that the Persian Gulf outflow waters could have played a role on your Holocene record. On that, perhaps you could be interested in reading the article by Naderi et al. (https://doi.org/10.1002/jqs.3614) could help having an illustration of what has been happening there and in the surrounding land.

**Response:** Thanks for the hint to the interesting new study about the postglacial flooding and Holocene climate shift in the Persian Gulf by Naderi et al. (2024), which also underlines our findings.

Finally, I am not sure the whole discussion on the wavelet analysis really adds a value to your discussion. I am not sure whether the sun is something had really a discernible impact on your SST record, given the other processes you list during the discussion, but I don't have strong recommendation to remove it either. Anyway listing the 525, 505, 493 etc. periodicities does not add something the reader will really focus on, and there is still the possibility that your sediment sampling could add wavelet artifacts on this long list of periodicities. I find it is a shame to discuss it, it dilutes your discussion.

**Response:** We understand, that the spectral- and wavelet analyses may not add a big value to our discussion part. We are also aware that the resolution can bias the analyses. Especially during the time period between 15 and 21 ka BP, the resolution of the SST reconstruction is not that high, due to the low organic content in this part of the core. However, also for other measured parameters, like $\delta^{15}N$ or total organic carbon (see Burdanowitz et al. 2024), we do not see strong changes during that period. In total, the findings of the spectral and wavelet analyses of the SST record are similar to the findings in Burdanowitz el al. (2024). However, we agree that a detailed discussion of this shifts the focus of our discussion. Therefore, we will condense this part in the discussion and delete the paragraph about the spectral and wavelet analyses in the conclusion.

Other minor remarks:

There is a series of typos (and bugs at reporting the chapter numbering). Pleaser get a profound last read over the manuscript prior to submitting your revised version.

**Response:** We will check the manuscript thoroughly for typos.

Productivity-mediated records in your core highlighted in Figure 4 could perhaps be better used while discussing the dynamics of your core.

**Response:** We will bring productivity reconstruction into the discussion in more detail and link SST reconstruction more closely with each other.

Thank you again for your time and effort. Your comments and suggestions helped us to improve our manuscript.

On behalf of all co-authors,

Jan Maier

---

## Author Comment (AC2)

**Comments Anonymous Referee #3:**

The study by Maier et al., "Spatial and temporal variability of sea surface temperatures and monsoon dynamics in the northwestern Arabian Sea during the last 43kyr" would be a significant contribution based on Alkenone proxy in the Arabian Sea.

There are only few studies based on Alkenone proxy in the study area and the present study with high resolution, long SST record is interesting work in the complex region and is accepted subject to minor modifications. There are few suggestions which authors need to address prior its acceptance.

Major comments

Gulf of Oman is a complex region and specifically influenced by SW monsoon due to its location and regional factors. Studying the region where seasonality is less pronounced due to the upwelling induced cooling during the summer season. Can the present study mark the variations in the seasonality of the two monsoon periods. Please elaborate on this aspect.

It is assumed that the alkenone proxy used in the study provides an Annual Mean SST. Does it not bias the study, can this give a true measure of the seasonality in the study area?

**Response:** First, we would like to thank the anonymous referee for her/his comments, which have helped us improve our manuscript.
We agree that while our SST data reflect an annual mean, the variations observed within the record still provide insight into seasonal changes. This is primarily due to the distinct influence of both the SW and NE monsoons on SST in our study area, particularly during their respective seasons.
Generally, we observe that stronger SW monsoon conditions prevail during interglacial periods, while glacial periods are marked by stronger NE monsoon conditions (Clemens et al., 1991; Prell and Kutzbach, 1992; Prell and van Campo, 1986). Other core locations in the Arabian Sea show a tendency to be more directly impacted by either the SW monsoon (e.g., MD00-2354) or the NE monsoon (93 KL, 136 KL). The unique location of our sediment core allows us to capture a more balanced view of these influences, where seasonal SST signals emerge based on glacial or interglacial periods. For instance, a warmer SST signal aligns with stronger SW monsoon conditions (as seen in D-O interstadials), while cooler SSTs are consistent with stronger NE monsoon activity (as during Heinrich event 4) or strong influence by the Oman Upwelling. We recognize that not every cold or warm event is captured in our record due to the complex interplay of other factors such as mesoscale eddies, SST gradients, northward expansion of the Oman Upwelling and NW winds. Nonetheless, the core's highly sensitive SST fluctuations, relative to other Arabian Sea locations, underlines its dependency to monsoon variability.

Annual SST variations are combination of different temperature signals such as Solar insolation and evaporative cooling, Strong NE winds, intensity of Upwelling, thus overcoming the bias by each signal would be difficult to estimate.

**Response:** We totally agree with the reviewer. This makes the core location very complex and difficult to interpret in terms of the individual drivers of SST changes. In combination with other proxies ($\delta^{15}N$) and other regional cores we tried to estimate what are the potential drivers behind the SST changes. However, given the complexity of the region, more research is needed to unravel the unsolved patterns.

Past studies based on alkenones and other proxies from Arabian Sea reveals cooling during the LGM which varied regionally.

"The Unusual SST pattern at LGM" is not explained properly, majorly all the records in the Arabian Sea suggest cooling of at least 2°C at LGM. There are several reasons as discussed in this section which prompts towards reduced SST contrary to what has been given as justification for warm SST at LGM, for example

1. (i)  When there is an intensified NE monsoon effect the associated SST would be lowered
2. (ii)  Reduced solar insolation compared to Holocene
3. (iii)  Weakened SW monsoon during the entire glacial period cited references such as Boll et al., 2015; Naidu and Malmgren, 2005; Schulte and Müller, 2001, also suggest reduced SST at LGM.
4. (iv)  Elevated dust levels could lower the SST then why there is a warm SST at the core site?

Infact, at the YD the low SST was explained with these reasons, compared to LGM's warm SST anomaly. All these points indicate towards lower SST's. Please provide proper justifications of warm SST at LGM.

**Response:**  We agree with the reviewer that typical SST patterns in the Arabian Sea during the LGM generally show cooling and all 4 remarks (i – iv) promote the cooling. In this chapter, we outline the key factors that typically led to SST drops during the LGM and then provide insights into why these factors may not have had the same impact in the Gulf of Oman:

As already mentioned in the chapter, do we need to keep in mind the unique, semi-enclosed location of Gulf of Oman. This may have retained heat more effectively compared to open Arabian Sea. Also, mesoscale eddy circulation and a shift of the SST gradient may also have retained heat more effectively in the Gulf of Oman, so that these processes probably outweighed the effects of the other four processes. Further, having a look at the absolute SSTs the differences between the northern Arabian Sea and the Oman Upwelling is quite low (between 0 and 0.5°C (Gaye et al. (2018)) but was a bit higher at the beginning and the end of the LGM (Gaye et al. 2018). Our record shows similar SSTs (around 26°C) at the beginning of the LGM as the 93KL in the northern Arabian Sea to drop down to almost 24°C around 19 ka with a smooth increase afterwards (Figure 4). The difference to other records from the Arabian Sea are the faster increasing SSTs right after its minima compared to other records. We will also clarify this point at the corresponding position in the manuscript.

Is there any time lag similar to what has been discussed by Naidu and Malmgren, 2005 or the chronology of the sediment core needs to tested for its robustness. Since in few studies the increase in sedimentation rate has been reported at LGM for various reasons in the basin.

**Response:** The chronology of the sediment core (covering the past 22 ka) studied by Naidu and Malmgren (2005) is based on thirteen AMS radiocarbon dates vs. twenty-one AMS radiocarbon dates (with sixteen in the similar time range of the studied by Naidu & Malmgren) of our sediment record (see Burdanowitz et al. 2024). We developed the age-depth model with the Bayesian model package BACON (v. 2.5.6) within R and using the Marine20 calibration curve and a deltaR of about 93+-61 years. Therefore, we are convinced that our age-model is robust. Further, in terms of solar insolation, we question the assumption that there is a SST time lag response of about 6 ka within the Holocene time period (around 12 ka duration so far). The sedimentation rates in core SL167 were relatively low during the LGM compared to the Pleistocene and Holocene (Burdanowitz et a. 2024).

The chronology part is discussed in Burdanowitz paper, however there should be an age model in the present study.

**Response: Response:** We agree with the referee (and the other two referees) that we need to discuss our age model in more detail. Therefore, we will add a paragraph and the figure of the age-depth model by Burdanowitz et al. 2024 into the material and methods section. We will also add the measured ages including their uncertainties as diamonds to figure 4.

Minor Comments
Line no. 62, d13O needs to be corrected to "d18O"

**Response:** We agree with the reviewer's comment and have revised the point accordingly and we will also check the manuscript thoroughly for more typos.

References should be in chronological order (for eg., line no. 72-75), this needs to be corrected throughout the manuscript. At several instances similar correction has to be made.

**Response:** We agree with the reviewer's comment and ensure that any additional errors in the chronological order of references are corrected.

Line no. 90, "500 -1500m", add space between the no. number and units. Similar comment for the line 96, 98, 99, then at 120 and at several other. This need to be corrected throughout the manuscript.

**Response:** We agree with the reviewer's comments and will check the hole manuscript for the correct spelling of number and units.

The initial sampling was done at 2 cm interval. However, the alkenone analysis has been done only for 219 samples. The reason stated behind this the concentration of organic matter. However, is there any difference in terms of resolution of the study at later half of the core?

**Response:** Analyses were conducted at 2 cm intervals down to a depth of 162 cm, and at 4 cm intervals below this depth due to lower organic carbon content (approximately <1.5%) and limited lipid material. This approach provides higher resolution data in the Holocene compared to the Pleistocene, while still maintaining a high level of resolution throughout the entire sediment core.

At line no. 199 "Enhanced SW monsoon conditions can also strongly impact the SST" If the SST is Annual average then how to distinguish the seasonality?

**Response:** While we assume that our SST data reflects an annual mean, variations within the record can still be indicative of seasonality due to the influence of the SW monsoon (and also the NE), which has a pronounced effect on SST, particularly in the summer month (or winter month for the NE monsoon). For example, stronger NE monsoon conditions during the LGM reflecting in general lower SST in northern Arabian Sea (e.g. 93 KL and 136 KL) and also in the Oman Upwelling Area (MD00-2354). Conversely, stronger SW monsoon winds after the LGM lead to a continuous warming in SST in the northern Arabian Sea (93 KL and 136 KL).

Why there isn't any trend in increase in SST at DO 2 and DO3?, Please provide justification for these two interstadials.

In general, for DO Interstadials strong NW monsoon was predicted, however the authors didn't mention about the effect of upwelling induced cooling on the SST in these time periods. Specifically looking at DO2 the low SST justification provided is due to intrusion of RSW or AIW in the Oman margin, with stronger mixing forced by NW/NE winds. However, at large at all the interstadials the SW monsoon winds were prevalent then what led to the change in contrasting wind conditions at DO2 interstadial? Please elaborate.

Is there any specific reason behind the increase in SST after DO2 and at the onset of LGM at 23 Ka.

**Response:** Our SST record demonstrate that the Gulf of Oman is highly sensitive to climate variations, responding readily to shift in both oceanographic and atmospheric conditions. However, these varying conditions can also retain or damp a pronounced SST signal. Since so many factors can influence the SST, individual warm or cold events may not be represented. This includes D-O 2 and and D-O 3 but also the LGM.

We also noted that the temperature increase around 23 ka may be attributed to the D-O 2 interstadial (following Böll et al., 2015), but our marking of the interstadial is a little earlier (ca 23-24 ka). However, we consider an increased influence from upwelling unlikely during this event, as a strongly pronounced OMZ is absent in our data (Figure 4f).

Instead, we observe an oxygen-depleted signal (Figure 4g), suggesting a more plausible influence from the intrusion of RSW or AIW rather than upwelling (following Burdanowitz et al., 2024).

At line no. 246, Moreover, during the LGM it has been postulated that there was an increased transport of warm water into GOM. The authors didn't provide any justification for the statement.

**Response:** Thank you for the suggestion. We intended to explain the transport of warmer water masses into the Gulf of Oman through mesoscale eddies. We will revise the sentence to clarify this point.

Global Factors influencing the SST variations at Gulf of Oman In this section, the present dataset compared with records of other regions show similar variations. However, at LGM variation doesn't match with global records and the authors suggest influence if regional processes. What significant factors change this trend when comparing the records at other time periods such as Heinrich events, DO and BA cycles, early Holocene etc. which doesn't apply compared to LGM?

**Response:** The observed discrepancy in SST variations during the LGM compared to other global records is indeed one of the key findings of our study. We hypothesize that regional processes such as a stronger NE monsoon conditions, intense NW winds and a shift in the SST gradient likely explain this discrepancy. However, we acknowledge that these conclusions are based on the current data set and further studies in the Gulf of Oman are needed to fully understand the underlying mechanisms and confirm these results. We will express this in our manuscript.

The spectral analysis periodicities of 7200 in the SST data which is attributed to Heinrich Events due to changes in Laurentide ice sheet. However, the authors may look at Naidu et al., 2019 wherein changes in the cyclicity is attributed to precision.

**Response:** The reviewer is right, that the 7200 year cycle can be also attributed to (subharmonic) the precession cycle as also mentioned by another reviewer. We will add and discuss this in our discussion.

Conclusion

The significant points of the present study should be elaborated rather just mere stating the results of the study.

In general the dataset is very interesting but the discussion lacks proper reasoning and interpretation and the authors are encouraged to touch upon the comments to provide a better insights on their study.

**Response:** Thank you again for your time and effort. Your comments and suggestions helped us to improve our manuscript.

On behalf of all co-authors,

Jan Maier

---

## Author Comment (AC3)

**Comments Anonymous Referee #2:**

In their manuscript Maier and colleagues present a new alkenone SST record from the Arabian Sea. Their record shows substantial millennial scale SST variability of 4C during both the last glacial and the Holocene, with the exception a quiescent period during the LGM. However, the glacial/interglacial offset is small, only around 1C. This result is in stark contrast to other Arabian Sea SST records which show large glacial/interglacial changes, and small millennial scale variability. This intriguing result potentially offers new insights into either the spatial variability of SSTs in the Arabian Sea and their sensitivities to elucidate different components of the climate system, or as a sensitive recorder of the Oman Upwelling Zone or more broadly the east-west SST gradient in the Arabian Sea.

While the paper provides good detail on specific changes in the record and potential mechanisms for these changes, it misses the big picture. The knowledge gap and research question are not stated and it's not clear what the jump is in our understanding of the regional climate system provided by this paper. It is hard to assess the validity of changes through time when the timing of SST changes in the presented record are rarely stated or given uncertainties. The lack of glacial/interglacial offset is not explored in detail. The impact of changes in bathymetry due to deglacial sea-level rise is only lightly touched upon. While individual Heinrich events are discussed, a common response is not, and no distinction is made between the response to Heinrich events versus Greenland stadials. I have further misgivings over the event chronology used as a comparison, which is based on Mediterranean cosmogenic exposure dates with external error of 4kyr, rather than using the INTIMATE chronology to focus on Greenland stadials and interstadials, or a U/Th dated speleothem which might be able to distinguish Greenland stadials and Heinrich events. The spectral analysis section is reasonable, but the focus is on what frequencies are found, when perhaps the most interesting result is a 5kyr window of no periodicity surrounding the LGM. Overall, this is an exciting record that lacks a quantified robust discussion. In its current form it is not ready for publication.

**Response:** First, we would like to thank the anonymous referee for her/his helpful and insightful comments that have contributed to improving the manuscript.
Our central motivation of this study is that there is a lack of SST data, particularly in the Gulf of Oman, which our study seeks to fill by focusing on the core location. Moreover, there are still few records with high-resolution SST reconstructions from the Arabian Sea. The core site in the Gulf of Oman is situated in an atmospheric (influenced by the SW and NE monsoon and NW winds) and oceanic (eddies, partly affected by the Oman Upwelling as well as inflow of Persian Gulf Water and strong SST gradients) dynamic region. Compared to other parts of the Arabian Sea, we show that northern hemispheric climate dynamics might be damped or strengthened by more regional processes in this complex dynamic region. Notably, we find that there is minimal cooling during the LGM compared to other locations in the Arabian Sea, and our site is more sensitive to temperature changes at millennial-scales than others. We can only speculate what are the dynamics behind the SST pattern at our core site and it asks for further paleoclimatic research in this part of the Arabian Sea. We will clarify the knowledge gap and research question, particularly in the abstract and conclusion.
It is correct that we do not pay much attention to the impact of changing water depth due to sea level rise. The core site is located at a water depth of about 770 m. Red Sea sea

level reconstructions by Siddal et al. (2003) and Rohling et al. (2008) show a sea level rise of about 80 to 100 m from the last glacial to the Holocene, which led to a connection between the Persian Gulf and the Arabian Sea. We do not expect a strong impact of sea level/water depth fluctuations on SST during the Pleistocene. However, the connection to the Persian Gulf since the postglacial sea level rise may indeed have influenced SST variations at our core site. The core site is located in the southern part of the Gulf of Oman and, therefore, in the area of PGW outflow. We will add further explanation to the discussion.

For the INTIMATE event chronology, Rasmussen et al. (2014) stated that the last about 44 ka are based on NGRIP and GRIP data sets: "*In the most recent extension of the INTIMATE event stratigraphy scheme, all events down to and including GI-12 were defined using NGRIP and GRIP δ18O data (Blockley et al., 2012). While most of the existing definitions are well supported by the [Ca2+] record, with δ18O shifts typically aligning with the early or middle parts of the corresponding [Ca2+] shifts, the [Ca2+] records support the choice of older onset points of stadials GS-9 and GS-10. However, yet again we retain current definitions to maintain compatibility with the existing event stratigraphy scheme.*" The reviewer is correct that the uncertainties of the Mediterranean cosmogenic exposure dates are comparably higher (about 4 ka) than those of the INTIMATE record (about 1 ka). However, firstly the timing and duration of most Heinrich events in our record are similar to those of Greenland Ice Core records. An exception is Heinrich 3 event, which is dated between 32.7 and 31.3 ka by Allard et al. (2021) and around 30.6 ka (Greenland Stadial (GS) 5.1 within H3 (Pedro et al. 2022)) and the GS 5.2 around 32.0 ka by the INTIMATE chronology (Rasmussen et al. 2014), with the latter falling in the timing of H3 by Allard et al. (2021). Secondly, the H3 event is followed by D-O 4. Fleitmann et al. (2009) have shown an age offset of the D-O 4 by about 590 years between the Sofular Cave in Turkey (earlier) and the NGRIP GICC05 chronology. Further, our δ15N record (Burdanowitz et al. 2024) matches the D-O timings of the more nearby Sofular Cave by Fleitmann et al. (2009) very well. For instance, the pronounced strong OMZ around 30 ka is associated with D-O 4, which would fall within H3 after the NGRIP records. Further, from the 21 AMS [14]C dates, two are around the timing of H3 with ages of 30250 +- 340 yr BP and 32590 +- 540 yr BP, respectively. Lastly, as discussed in Burdanowitz et al. (2024), some of the existing records in the Arabian Sea are tuned to the NGRIP ice cores, and the peaks then of course fit in the timing of the Heinrich and D-O events. Altogether, we are convinced that our age model is robust and independent, and the observed age offsets between the NGRIP cores and the more regional cores are not due to age uncertainties. Therefore, we will stick to the Allard et al. (2021) timings of the events. Further we will stick with the nomenclature of Dansgaard-Oeschger events to be consistent with our earlier findings and as they are used according to the Greenland Interstadials (e.g. Rasmussen et al. 2014)

We will provide specific responses to each of the line-by-line comments in your specific comments section.

**Specific Comments**

Line 20 states that the dynamics of monsoon periods are influenced by northern hemisphere solar radiation, but the paper is largely about Heinrich events and Dansgaard-Oeschger interstadials and other abrupt millennial scale events, not solar radiation.

**Response:** Thank you for your insightful comment regarding the influence of northern hemisphere solar radiation on monsoon dynamics. We recognize that the primary focus of our paper is on Heinrich events, Dansgaard-Oeschger interstadials, and other abrupt millennial-scale events as they overprint the general influence of the solar radiation. However, we included the discussion of northern hemisphere solar radiation to provide a broader context for understanding the relevant climatic mechanisms during these periods.

Knowledge Gap/Research Question: It's not clear what the purpose of this study is. What is it trying to find out? At the paragraph break line 51-52 we need to understand what is the unknown which this study intends to investigate. A one or two sentence summary of Gaye et al. 2018 of key results might be useful, as this might outline what the knowledge gap is.

**Response:** We will clarify the purpose of our study giving a brief summary of the key results from Gaye et al. (2018) covering the past about 25 ka. They found that the glacial SST were about 4°C lower than SST's during the Holocene in the Arabian Sea. Further they postulate that the general glacial SST gradient within the Arabian Sea had a stronger N-S insolation driven component than during the Holocene with a more pronounced NW-SW circulation driven component. We will also ensure that the knowledge gap and the purpose of our study are explicitly highlighted in both the abstract and conclusion sections to reinforce their significance.

Age model uncertainty: Dating uncertainty and age model uncertainty need to be stated in section 3: i.e. a key summary of the Burdanowitz et al., 2024 age model.

**Response:** We agree with the referee (and the other two referees) that we need to discuss our age model in more detail. Therefore, we will add a paragraph and the figure of the age-depth model by Burdanowitz et al. (2024) in the material and methods section. We will also add the measured ages including their uncertainties as diamonds to figure 4.

Greenland stadials vs Heinrich events: This paper uses the nomenclature of Dansgaard-Oeschger interstadials and Heinrich events. It is not clear why Heinrich events in particular are different from Greenland stadials in this record. I suggest switching to a Greenland stadial nomenclature throughout the manuscript. This would also allow for more precise timings from the INTIMATE chronology to be used in preference to the Mediterranean cosmogenic dataset currently used.

**Response:** As stated earlier, the INTIMATE chronology is based on NGRIP and GRIP data sets. Further, some of the Greenland stadials are Heinrich events (e.g., GS 5.1 is related to H3 (Pedro et al. 2023)). As mentioned above it is correct that the uncertainties of the Mediterranean cosmogenic exposure dates are comparably higher (about 4 ka) than those of the INTIMATE record (about 1 ka). However, firstly most of the timing and duration of the Heinrich events are similar to those of Greenland Ice Core records, except H3. Secondly, the

H3 event is followed by D-O 4. Fleitmann et al. (2009) have shown an age offset of the D-O 4 by about 590 years between the Sofular Cave in Turkey (earlier) and the NGRIP GICC05 chronology. Further, our $\delta^{15}N$ record (Burdanowitz et al. 2024) matches the D-O timings of the more nearby Sofular Cave by Fleitmann et al. (2009) very well. For instance, the pronounced strong OMZ around 30 ka is associated with D-O 4, which would fall within H3 after the NGRIP/INTIMATE chronology. Further, from the 21 AMS $^{14}C$ dates, two are around the timing of H3 with ages of 30250 +- 340 yr BP and 32590 +- 540 yr BP, respectively. Lastly, as discussed in Burdanowitz et al. (2024), some of the existing records in the Arabian Sea are tuned to the NGRIP ice cores, which peaks then of course fit in the timing of the Heinrich and D-O events. Altogether, we are convinced that our age model is robust and the observed age offsets between the NGRIP cores and the more regional cores are not due to age uncertainties. Therefore, we will stick with the Allard et al. timings of the events. Further we will stick with the nomenclature of Dansgaard-Oeschger events to be consistent with our earlier findings and as they are used equivalent to the Greenland Interstadials (e.g. Rasmussen et al. 2014).

The timings of the shaded bars in figures 4 and 5 are inaccurate. The 4.2 ka event is not 5-4 kyr BP. The 8.2 ka event is not 9-8 kyr BP. The YD is not 13-12 kyr BP. H1 is not 18-15.5. H2 is tricky to tie down, but I think it is more likely to be where DO-2 is currently labelled (this SST drop would help the case being made that Heinrich events are associated with SST decreases). I believe H3 is more likely to be GS5.1 than GS5.2. I don't know how much of this is a plotting issue and how much is an issue with the choice of external event chronology.

**Response:** Thank you for your detailed observations regarding the timings of the shaded bars in Figures 4 and 5. We intentionally designed the bars to be larger for visibility, as smaller bars would be difficult to discern. Additionally, we acknowledge that events can have varying onset times, and thus we have allowed for some flexibility in their representation. The designations for glacial and interglacial periods have been carefully chosen and align with those established by Burdanowitz et al. (2024), ensuring accuracy in our representation. Further, we disagree that a decrease of SST is strictly associated with Heinrich Events as the study is highly complex which needs to be taken into account. Only the H4 event led to a significantly drop of SST's at the core site. As stated above, we are convinced that our independent age model is robust and we will not tune it to get a "better fit" to the NGRIP/INTIMATE chronology. We clearly have no signal of GS 5.1 in our SST record and our d15N record (Burdanowitz et al. (2024)) shows at the timing of GS 5.1 (after the INTIMATE chronology) indications of the D-O 4, which matches with the cave record of Fleitmann et al. (2009).

Glacial/Interglacial change: The lack of a large change in SST from glacial to interglacial (about 24.5 to 25.5 C by my eye) compared to nearby records (2-4 C) is one of the remarkable results of this paper. Yet it is not discussed. The role of changing bathymetry is similarly only lightly discussed.

**Response:** The most significant impact of sea-level change occurred at the end of the Pleistocene leading to the flooding of the Persian Gulf, which we discussed in detail. For smaller sea-level changes, the influence of water depth changes becomes less critical,

as these variations do not significantly affect SST unless they result in substantial alterations of the ocean circulation (Rohling et al., 2008; Siddall et al., 2003). The core site is located in an area with several atmospheric and oceanic influences. Firstly, the LGM has not a strong impact on SST at the core site. Secondly, during the Holocene, the SST variations are strong but lying between SST's in the northern/northeastern Arabian Sea and the Oman Upwelling region. This shows that the core site is affected by both the NE and the SW monsoon. Further, the connection to the Persian Gulf became important during the Holocene. Lastly, the strong SST gradient, which is also visible in modern SST records, within the Gulf of Oman have a strong impact on SST reconstruction as a small shift can have a large impact (see figure 2). However, we will point out the lack of the glacial/interglacial changes more clearly in the discussion.

Mid-Holocene cooling: What is the timing of the SST decrease at 5ka relative to the end of the Green Sahara? And does this have any mechanistic implications?

**Response:** Thank you for your inquiry regarding the timing of the SST decrease at 5 ka in relation to the end of the Green Sahara. We agree that the SST decrease around 5 ka coincides closely with the transition out of the Green Sahara period. This timing suggests a potential link between the cessation of the Green Sahara and regional climate dynamics, as shifts in monsoon intensity, changes in atmospheric/oceanic circulation patterns may have contributed to observed SST cooling in our record. We will discuss this in more detail in the discussion.

2 ka event: I am skeptical here without a detailed description of timing and uncertainty. A climate anomaly between 5 and 4 kyr BP is no longer sufficient to be labelled as the 4.2 ka event. It needs to be coincident with the Carolin et al., (2016) window of 4.26 to 3.97 kyr BP. I think the discussion of the event starting at line 312 is reasonable. But line 316 goes against the previous discussion. I would remove references to 4.2 from elsewhere in the paper (such as the results and conclusions) but keep the brief discussion around 312 as this part is nuanced and reasonable. Unless of course, specific data can be provided to support the assertions made.

**Response:** Thank you for your comments regarding the 4.2 ka events and its timing. As previously mentioned, we used the broader time frame of 4–5 ka for clarity in our figure, as a more specific delineation may not be easily discernible. While we acknowledge the time frame from Carolin et al. (2016), we believe that uncertainties and potential shifts in the timing of events can lead to variations, making it plausible for the 4.2 ka event to occur slightly earlier or later. Therefore, we would like to retain references to both the 8.2 ka and 4.2 ka events in the manuscript.
We appreciate your observation regarding the sentence in line 316, which appears contradictory to the preceding discussion. We will revise this sentence to ensure consistency and clarity throughout the text. Thank you again for your constructive feedback, which will help to improve the manuscript.

Spectral Analysis: There is a significant reduction in cyclicity between 22 and 17kyr BP (ish). This quiescent LGM might warrant further investigation.

**Response:** Thank you for your insightful observation regarding the significant reduction in cyclicity between approximately 22 and 17 ka BP during the Last Glacial Maximum (LGM). We totally agree and acknowledge that this stable period represents an intriguing aspect of our data that may indeed warrant further investigation.

The 7200 year cycle is a potential subharmonic of precession, modulated by obliquity.

Might the high number of potential cyclicity peaks around 500 years be one periodicity on top of an uncertain chronology?

**Response:** We agree that the 7200 year cycle might be a subharmonic of the precession. We will add this potential mechanism to the discussion part. Regarding the high number of potential cyclicity peaks around 500 years, we recognize that these could be indicative of a true periodicity. However, we also acknowledge that they may reflect the uncertainties inherent in the chronology of the data. The uncertainties of our age model are between 170 to 240 years for the Holocene and up to 770 years for the oldest part of the record.

**Technical Corrections**

Line 29: insert modern: "of the total modern annual precipitation"

**Response:** We agree with the reviewer's comment and have revised the point accordingly.

Line 41: "Northern Hemisphere glacial ice-sheets"

**Response:** We agree with the reviewer's comment and have revised the point accordingly.

Line 48: no need for a comma between 'both' and 'the'.

**Response:** We agree with the reviewer's comment and have revised the point accordingly.

Line 49: "in the past" is redundant and could be deleted.

**Response:** We agree with the reviewer's comment and have revised the point accordingly.

Line 65: The 'Study Area' section is more of a 'Modern Climate Dynamics' section.

**Response:** We agree with the reviewer's comment and have revised the point accordingly.

Line 93: New Paragraph?

**Response:** We agree with the reviewer's comment and have revised the point accordingly.

Line 119: This sentence is a bit confusing. I suggest 'Alkenones were measured at 2cm for the upper 162cm and 4cm resolution below 162cm by combining consecutive subsamples due to lower organic content'.

**Response:** We agree with the reviewer's comment and have revised the point accordingly.

Line 120: suggest '3 to 18g of sediment'.

**Response:** We agree with the reviewer's comment and have revised the point accordingly.

Line 122: space needed between sentences.

**Response:** We agree with the reviewer's comment and have revised the point accordingly.

Line 123: This sentence is out of place. Methods should be written in chronological order.

**Response:** We agree with the reviewer's comment and have revised the point accordingly.

Line 144: This is precision not accuracy

**Response:** We agree with the reviewer's comment and have revised the point accordingly.

Line 164: Figure reference needed here.

**Response:** We agree with the reviewer's comment and have revised the point accordingly.

Line 187: 'mitigated' is not correct here. Is 'reduced' sufficient?

**Response:** We agree with the reviewer's comment and have revised the point accordingly.

Line 214: 'westerlies'

**Response:** We agree with the reviewer's comment and have revised the point accordingly.

Line 267: suggest 'minor decrease in Central Arabia'

**Response:** We agree with the reviewer's comment and have revised the point accordingly.

Line 308: New paragraph.

**Response:** We agree with the reviewer's comment and have revised the point accordingly.

Line 320: period missing at the end of the sentence.

**Response:** We agree with the reviewer's comment and have revised the point accordingly.

Line 338: 'Implies' is incorrect here. Suggests or is linked to might be better.

**Response:** We agree with the reviewer's comment and have revised the point accordingly.

Line 80: This sentence might need a rewrite to make it clearer. At present (4-5C) as a quantification of several hundred kilometers makes no sense.

**Response:** We will revise this sentence.

Line 65-85: Better disambiguation between spatial and temporal SST gradients is needed.

**Response:** We will correct the sentence as followed: A spatial SST gradient of 4-5 °C develops over several hundred kilometers with the onset of the SW monsoon, creating a temperature low near the coast of Oman and a high of approximately 29 °C in the western Gulf of Oman (Figure 2c).
We are referring specifically to the spatial SST gradient generated by the SW monsoon onset, with cooler temperatures near the coast of Oman and warmer temperatures in the western Gulf of Oman. We did not refer to the temporal SST gradient here. We understand that this may have caused confusion due to the title, which includes both spatial and temporal SST variability.

Line 101: Need a linking sentence or signpost sentence to help the read transfer into this new paragraph.

**Response:** We concur with the reviewer´s observation and updated the sentence: Building on complex interaction within the Arabian Sea, mesoscale eddies are cyclonic and anticyclonic rotating water masses, contrary to the surrounding main currents and emerge as key players in the regulation of surface ocean circulation (Fischer et al., 2002; de Marez et al., 2019; Al Saafani et al., 2007; Trott et al., 2019).

Line 151: The resolution of the evenly spaced dataset needs to be stated alongside the resolution of the actual data, so that this choice can be evaluated.

**Response:** We will add the missing information to the material & methods part. As mentioned in the manuscript we are using the package ncdf4.helpers v.0.3-6 (Bronough, 2021) and the approx. function. We used the highest resolution of the record by using the function "get.f.step.size()" resulting into a minimum step size of 40 years between two measurements. With that we were able to define the amount of time steps (in total 890) to generate an even spaced dataset. We are aware that this may lead to uncertainties, especially for the parts of the core with a lower resolution. The resolution of the original data sets vary between 40 and 800 years (mean: 181 +- 124 years, median: 168 years), with lowest resolution during the LGM.

Line 250: 17ka is not earlier than 19ka

**Response:** In this case, "earlier" response not to 19 ka at the beginning of the paragraph but to the B-A around 14 ka in the sentence before the usage of "earlier". Following the usage of "earlier around 17 ka" is correct.

Line 258: its not clear what is meant by the NW winds moving in the opposite direction to the SW monsoon.

**Response:** We will rewrite the sentence to make it more clear: NW winds, peaking between 15 and 13 ka (Sirocko et al., 2000), may have contributed to the earlier warming, while SW monsoon conditions weakened (Leuschner and Sirocko, 2000; Sirocko et al., 2000).

Line 299: Lead with the evidence, then the interpretation, not the other way round.

**Response:** We will correct the sentence as followed: With the beginning of the mid-Holocene, strengthening of NE monsoon conditions likely led to a temporarily interrupted transport of upwelled water masses to the core location.

Line 337: D-O interstadial?

**Response:** The intention of our sentence was to illustrate the contrasting effects associated with different climatic events, such as a weakening (or strengthening) of the AMOC during the 8.2 ka cold event compared to D-O interstadials. We aimed to highlight the opposing shifts of the ITCZ and the corresponding impacts on the ISM. We will ensure this contrast is more explicitly stated in the text for clarity.

Thank you again for your time and effort. Your comments and suggestions helped us to improve our manuscript.

On behalf of all co-authors,

Jan Maier